



# Impact of increased resolution on the representation of the Canary upwelling system in climate models

Adama Sylla [1,2], Emilia Sanchez Gomez[1], Juliette Mignot[3], and Jorge López-Parages[1,4]

[1]CECI, Université de Toulouse, CNRS, CERFACS, Toulouse, France
[2]Laboratoire de Physique de l'Atmosphère et de l'Ocean Simeon Fongang (LPAO-SF/ESP/UCAD), Dakar, Senegal
[3]Sorbonne University (CNRS/IRD/MNHN), LOCEAN Laboratory, Paris, France
[4]Physical Oceanography Group, University of Málaga, Spain.

**Correspondence:** Adama Sylla (adamasylla36@gmail.com)

**Abstract.**

We investigate the representation of the Canary upwelling system (CUS) in six global coupled climate models operating at high and standard resolution as part of the High Resolution Model Intercomparison Project (HighResMIP). For this project the resolution of the ocean and/or atmosphere components was increased. The models performance in reproducing the observed

CUS is assessed in terms of various upwelling indices based on SST, wind stress and sea surface height, focussing on the effect of increasing model spatial resolution. Our analysis shows that an increase of spatial resolution depends on the sub-domain of the CUS considered. Strikingly, along the Iberian Peninsula region, which is the northernmost part of the CUS, the models show lower skill at higher resolution compared to their corresponding lower resolution version in both components for all the indices analyzed in this study. On the contrary, over the southernmost part of the CUS, from the north of Morocco to the Senegalese

coast, the high ocean and atmosphere resolution models simulate a more realistic upwelling than the standard resolution models, which largely differ from the range of observational estimates. These results suggest that increasing resolution is not a sufficient condition to obtain a systematic improvement in the simulation of the upwelling phenomena as represented by the indices considered here, and other model improvements notably in terms of the physical parameterizations may also play a role.

## 1 Introduction

The upwelling is an upward motion of sea water from intermediate depths toward the ocean surface resulting from the friction of the surface wind on the ocean surface. Upwelled water masses are colder and richer in nutrients than the surface waters they replace. Therefore, upwelling zones correspond to areas of very productive marine ecosystems and high fish resources. In for areas where the upwelling occurs along the coast, this phenomenon presents a noticeable socio-economic importance or

the countries concerned, in particular in relation to the fisheries sector (Gómez-Gesteira et al., 2008). From the physical point of view, coastal upwelling is mainly caused by prevailing trade winds blowing equator-ward parallel to the coastline, which push the surface waters away the coast through the so-called Ekman transport. As a result, the divergent flow at the surface is compensated by an onshore flow from below that brings colder and nutrient-rich waters to the surface. In addition, positive





divergent oceanic circulation may also be triggered at the surface by a cyclonic wind stress curl. Indeed, in the eastern sub-
tropical basins, where trade winds tend to slow down near the coast, the wind drop-off induces a positive wind stress curl that
also contributes to upwelling (Pickett, 2003 and Bravo et al., 2016). This second effect is associated with the so-called Ekman
pumping, which acts to side up deeper waters into the euphotic zone. Both offshore Ekman transport and Ekman pumping
contribute to enhance the nutrients levels in leading to an enhanced biological production along the coast (Pennington et al.,
2006). There are four major coastal upwelling systems (hereafter EBUSs for Eastern Boundary Upwelling Systems) in the
global ocean, that are the Canary, Benguela, Humboldt and California systems. These areas cover less than 1% of the global
ocean surface, but they contribute more than 20% of the global fish catches (Pauly and Christensen, 1995).

Among the four EBUSs mentioned above, we focus here on the Canary Upwelling System (CUS), which extends from the
northern tip of the Iberian Peninsula at 43°N to the south of Senegal at about 10°N (Fig.1). The variability of this upwelling
system has been studied on time scales ranging from synoptic to seasonal (Torres, 2003 and Alvarez et al., 2005). It has also
been studied on longer timescales, but to a lesser extent, due to the lack of sufficiently long, continuous time series (Blanton
et al., 1987). In the CUS, the strength of the upwelling favorable winds are associated with latitudinal variation of the Inter-
tropical Convergence Zone (ITCZ) and the Azores high pressure system. The latter, which is part of the Hadley-circulation
latitudinally migrates from 25°N in late winter and 35°N in late summer. The Azores High drives both the intensity and the
latitudinal extension of the north-easterly winds along the CUS.

According to previous studies, the CUS can be divided into different sub-systems based on its circulation, physical envi-
ronment and shelf dynamics ( Santos et al., 2005; Gómez-Gesteira et al., 2008 and Arístegui et al., 2009). The western coast
of the Iberian Peninsula located between 37°N to 43°N (hereinafter IP), is the northern limit of the CUS. The IP presents a
discontinuity in the flow with the northwest African coast (Arístegui et al., 2004). This is caused by the presence of the Strait
of Gibraltar, which allows the exchange of water between the Mediterranean Sea and the Atlantic Ocean. Upwelling activity
along the western coast of the IP occurs during boreal summer due to the poleward migration of the Azores high, which leads
to northerly winds flowing along the coast (Wooster et al., 1976; Fraga, 1981; Blanton et al., 1984; Bakun and Nelson, 1991;
Gomez-Gesteira et al., 2006; deCastro et al., 2008a; Alvarez et al., 2008 and Pires et al., 2013). Furthermore, the narrow shelves
of IP coast result in lower annual biological productivity than in the other sub-regions of the CUS (Arístegui et al., 2009). In
the area surrounding the Gibraltar Strait (from latitude 33°N to 36°N), the upwelling is drastically reduced.

The Morocco upwelling system (hereinafter MoUS), located form 21°N to 32°N is the central part of the CUS. According
to several studies the MoUS can be divided into two sub-domains: the northern part (nMoUS) and the southern part (sMoUS),
both extending between 25°N and 32°N and 21°N and 25°N (Santos et al., 2005; Gómez-Gesteira et al., 2008 and Arístegui
et al., 2009). In the nMoUS, upwelling occurs during the boreal summer, while the sMoUS is one of the few locations in the
world where upwelling is persistent throughout the year. This permanent upwelling is due by the fact that, unlike in the case of
the other sub-regions within CUS, in sMoUS the prevailing winds are always parallel to the coastal line.





Finally, in the southern part of CUS, the Senegalo-Mauritanian Upwelling System (SMUS), which extends from 12°N to 20°N
is the southernmost part of the CUS. Here the upwelling occurs from November to May, when the ITCZ reaches its southern-
most position (Faye et al., 2015 and Sylla et al., 2019).

In the last decades, the sensitivity of EBUSs to climate change has received increasing attention (Bakun, 1990; McGregor
et al., 2007 and Barton et al., 2013). Improving our knowledge of the response of the CUS to global warming is of crucial
importance since the food resources and economy of neighbors countries greatly depends on its characteristics and evolution
in the coming decades. Bakun (1990) suggested that coastal upwelling intensification would occur in response to continued
global warming. Specifically, he argued that anthropogenic climate change air temperatures on the continents are expected to
rise more than in the adjacent oceans (Manabe et al., 1991), producing a deepening of the thermal low-pressure systems over
land which lead to an intensification of the land-sea sea level pressure gradients, and a subsequent increase of summertime
upwelling-favorable winds (Rykaczewski et al., 2015). Efforts to test Bakun's hypothesis of upwelling intensification under
the recent warming trend are challenged by the limited spatial and temporal extent of observations (Cardone et al., 1990). In
this context, climate models offer an alternative method to simulate large-scale representation of the CUS and its sensitivity
to increased greenhouse gas concentration. By using upwelling indices based on wind-stress and/or Sea Surface Tempera-
tures (SSTs), Wang et al. (2015) and Sylla et al. (2019) show that climate models, as those participating within the Coupled
Model Intercomparison Phase (CMIP) exercises, are able to capture the main characteristics of the EBUSs. According to Pick-
ett (2003), the success of these low-resolution estimates of coastal upwelling may depend on their implicit integration of both
near-shore Ekman transport and offshore Ekman pumping. Nevertheless, standard resolution global climate models suffer from
several limitations as they do not represent finer scale processes associated with the upwelling, in particular the structure of
the offshore wind-stress divergence and curl (Small et al., 2015 and Vazquez et al., 2019 and references therein). Indeed, most
coupled climate models incorrectly simulate various processes occurring in these regions and suffer from warm SST biases
(Richter, 2015). Misrepresentation of stratocumulus clouds has been identified as one of the primary reasons for this bias.
Model inability to produce stratocumulus decks can lead to absorption of excessive shortwave radiation in the upper-ocean and
anomalously warm SSTs, which in turn induces a positive feedback to the initial error (Richter, 2015 and Zuidema et al., 2016).
Further, model resolution and errors in surface winds could also play a role through their impact on turbulent fluxes, coastal
upwelling and offshore Ekman transport (Gent et al., 2010; Richter, 2015; Zuidema et al., 2016 and Ma et al., 2019). Most
coupled climate models suffer from warm biaises, in SST and wind, near the subtropical eastern boundary regions (Davey
et al., 2002; Richter and Xie, 2008 and Richter et al., 2012). Despite numerous improvements in models over time, these
problems have persisted because of their coarser resolution. However increasing model resolution leads to an improvement of
the upwelling representation, in particular the SST warm biases over the upwelling regions are reduced (Small et al., 2015).
Some works have concluded that model resolution influences on the the overall representation of the mean climate over the
Tropical Atlantic (Doi et al., 2012 and Exarchou et al., 2018) and on the tropical north Atlantic response to remote forcings
such as ENSO (López-Parages and Terray, 2022). In line with these studies, Gent et al. (2010) report that an increase of the
nominal resolution of the atmospheric grid from 2° to 0.5° lowers SST biases up to 60% within the major upwelling areas.





Harlass et al. (2015) found that a significant improvement of warm biases within the Tropical Atlantic can be reached with a
simultaneous refinement of the horizontal and vertical resolutions of the atmospheric grid. Small et al. (2015) claimed that a
good representation of the upwelling systems as the Benguela (coastal upwelling along the southern African coast) requires an
eddy-resolving ocean model and an atmospheric model with enough resolution ($\sim 0.5°$) to capture realistically the wind stress
curl over the eastern boundary of the Tropical Atlantic Ocean.

Thus in the last few years, modelling centers have made a great effort to develop higher resolution global climate models.
The recent CMIP6 exercise coordinated the High Resolution Model Inter-comparison Project (HighResMIP/PRIMAVERA),
aiming to assess the benefits of increased resolution in climate models (Haarsma et al., 2016). Climate models resolution was
drastically increased in the atmospheric and oceanic components and for the first time a coordinated protocol was proposed
to asses the impact of enhanced model resolution in the representation of the climate system. This is a topic of growing in-
terest, particularly as some recent simulations suggest improvements in both large-scale aspects of the atmospheric and ocean
circulation and in small-scale processes and climate extremes (Haarsma et al., 2016; Roberts et al., 2016; Vanniere et al., 2019
and Hewitt et al., 2020). So far, the effects of increased CMIP6 model resolution on the upwelling systems has still not been
assessed. Here, we provide the first detailed analysis of the potential benefits of increasing model resolution in simulating the
CUS. To this aim, we compare the model performance in representing the upwelling indices defined in Sylla et al. (2019) by
using the standard and high resolution versions of some of the climate models that participated within the CMIP6 HighResMIP
project. The upwelling indices used in this study are based on SST and wind stress. The SST index aims at describing the
surface thermal signature of the CUS upwelling. Although this is a simplified view of the upwelling, this index has the advan-
tage of being based on a well-observed variable so that it can be properly constrained by observations. The other three indices
used here are based on the surface wind stress and meridional gradients of sea level. They aim at quantifying key mechanisms
implicated in the generation of upwelling vertical velocities: coastal divergence of the Ekman transport, Ekman pumping, and
possible counteracting effects due to convergences of the geostrophic flow.

The manuscript is organized as follows: Section 2 describes the numerical experiments, the observational and reanalyses
datasets as well as the different metrics used in this study. Section 3 provides a characterization of the upwelling in observations,
while the role of the model resolution in simulating the CUS is assessed in Section 4. Finally, results are discussed and a
conclusion is provided in section 5.

## 2   Data and methodology

### 2.1   Models and numerical experiments

The coupled models considered in this work are those participating in the European H2020 HighResMIP/PRIMAVERA project
(https://www.primavera-h2020.eu) which is part of the international HighResMIP exercise. We use the outputs of coupled
historical experiments (referred to as 'hist-1950') covering the period 1950-2014. In particular, the models used are HadGEM3





GC3.1 (Williams et al., 2018), CNRM-CM6-1 (Voldoire et al., 2019), CMCC-CM2 (Cherchi et al. 2019), MPI- ESM1 (Gutjahr et al., 2019), ECMWF-IFS (Roberts et al., 2018), and EC-Earth3P (Haarsma et al., 2020). The main characteristics of these models are listed in Table 1 together with the respective effective resolutions in both the atmosphere and ocean components.

Note that for the PRIMAVERA coordinated project, resolution was increased in both the atmosphere and the ocean components, with the exception of CMCC-CM2 and MPI-ESM1, in which only the atmospheric resolution was modified. Based on the change of ocean and/or atmosphere resolution four groups of models are defined: groups 1 and 1* including low resolution models (LR), and groups 2 and 2* including high resolution models (HR) for both the atmosphere and the ocean. From group 1 to group 2, both the ocean and the atmosphere resolution is increased. From group 1* to group 2*, only the atmosphere

resolution is increased. Note that our set of model is an ensemble which does not allow a precise comparison of the effect of increasing both ocean and atmosphere resolution on the one hand (groups 1-2), or only increasing the atmosphere resolution on the other hand (1*-2*). Indeed, the resultant model groups do not contain the same models and are not of the same side. Thus, some of the differences among the ensembles of model groups may be due to the intrinsic models biases rather than an effect of model resolution. This drawback has to be kept in mind.

Our analysis is based on the SST wind stress, sea surface height and mixed layer depth monthly fields (see section 2.3). For each variable we compute the 30 years climatological mean for the period 1985-2014. The choice of this period is motivated by the selection of a common period among the various observational datasets used in this study. To avoid biased multi-model ensemble means, only one member of each model was used even if several members are available for certain models. Note that the different members have been averaged together in order to increase robustness of the seasonal cycle estimation, nevertheless

the results does not change (not shown). Additionally, the choice can be also justified by the fact that our metrics are based on climatological averages and not on variance or trends metrics, which are more sensitive to internal climate variability.

## 2.2    Observational and reanalyses products

Several observational and reanalysis datasets are used in the present analysis (see Table 2 for details) in order to evaluate model results realism in simulating the CUS. For SST, we use the monthly HadISST.2 dataset, which was developed at the Met

Office Hadley Centre for Climate Prediction and Research (Titchner and Rayner, 2014) We have also used the version 2 of the National Oceanic and Atmospheric Administration (NOAA) Optimum Interpolation Sea Surface Temperature (OISST-v2) analysis (Reynolds et al., 2007). The OISST analysis combines Advanced Very High Resolution Radiometer (AVHRR) satellite data, buoy and ship-based observations from the International Comprehensive Ocean Atmosphere Data Set (ICOADS) database (Worley et al., 2005) . Although these data are provided with at daily frequency, monthly averages have been computed. Finally,

we have also included the latest version of the Extended Reconstructed Sea Surface Temperature data set (ERSST-v5, Huang et al., 2017). The monthly ERSST-v5, produced by the National Oceanic and Atmospheric Administration (NOAA), is based on in situ (ship and buoy) observations from ICOADS.

The Quikscat wind speed and the zonal and meridional components of the 10-m wind from the Cross-Calibrated Multi-

Platform (CCMP) project are also analyzed (Freilich and Spencer, 1994; Atlas et al., 2011). In addition to the observational





products, near surface wind data from two atmospheric reanalysis have been considered and compared to the previous observational winds products using a wind rose diagram (Fig A1 in Appendix A): the latest climate reanalysis ERA5, provided by the European Centre for Medium-Range Weather Forecasts (ECMWF; Hersbach et al., 2020). The latter is supported by the version 3 of the NOAA-CIRES-DOE 20th Century Reanalysis (20CRv3) Project (Slivinski et al., 2019). The NOAA-20CR-v3

datasets are supported by the National Oceanic and Atmospheric Administration (NOAA), the Cooperative Institute for Research in Environmental Sciences (CIRES), and the U.S. Department of Energy.

Because of a lack of wind stress observations and reanalysis covering the entire domain and period of our study (1985-2014), the surface wind speed is converted into wind stress following an empirical method (see Appendix A for more details). Note that this offline computation of the wind stress is only performed for the observations and reanalyses wind datasets but not for

the model, which directly provides the wind stress field.

Meridional sea surface height (SSH) gradients may also play an important dynamical role in coastal regions through geostrophic transport (Marchesiello and Estrade, 2010). Cross-shore geostrophic transport can substantially alter the vertical transport relative to wind-based estimates ( Rossi et al., 2013 and Jacox et al., 2014). Thus including the geostrophic

component is also important to assess the realism of the modeled upwelling (Rykaczewski et al., 2015 and Oerder et al., 2015). To evaluate the models representation of SSH along the CUS, we use the AVISO satellite altimetry product (Ducet et al., 2000). For comparison, we have also used the monthly mean SSH from GODAS. Furthermore, to quantify the effect of the SSH gradient on the geostrophic transport requires an estimation of the oceanic mixed layer depth (MLD). We use the MLD climatology from de Boyer Montégut, 2004. Monthly climatologies over the 1985-2014 period are considered for all the validation datasets

and models as specified in section 2.1, except for CCMP and AVISO, which are both based on a shorter time period (1992-2011 and 1995-2005 respectively).

## 2.3 Upwelling indices

We compute the upwelling indices developed in Sylla et al. (2019) for the SMUS and applied here to the whole CUS. The relevance of these indices to represent CUS variability is justified in section 3. We consider the SST difference between the

coast (black dots all region, Fig. 1) and the outer ocean (magenta dots, Fig. 1) in such a way that the SST-based index is defined by:

$$UI^{sst} = SST_{ocean} - SST_{coast} \tag{1}$$

Usually a distance of $5°$ longitude from the coast is considered for this index (Cropper et al., 2014 and Sylla et al., 2019). This SST upwelling index has been widely used to characterize upwelling intensity as it measures the impact of upwelling on the

SST zonal structure (Mittelstaedt, 1991; Santos et al., 2005; Gómez-Gesteira et al., 2008; Lathuilière et al., 2008 and Marcello et al., 2011). Positive (negative) values of the index correspond to more intense upwelling (downwelling).





As described in the introduction section, the action of the wind in the upwelling can be separated into two mechanisms (Sverdrup et al., 1942; Yoshida, 1995 and Smith, 1968). The first mechanism, called the Cross-Shore Ekman Transport (here-

after $CSET$) is commonly used for characterizing coastal upwelling (Bakun, 1973; Schwing et al., 1996 and Gómez-Gesteira et al., 2008). $CSET$ is computed as the offshore component of Ekman transport ($Q$), whose zonal and meridional components are derived from the wind stress field as follows:

$$Q_x = \frac{\tau_y}{\rho_w f} \quad and \quad Q_y = -\frac{\tau_x}{\rho_w f} \tag{2}$$

where $\rho_w$ is the sea water density (1025 kg.$m^{-3}$) and $f$ is the Coriolis parameter. Following Santos et al. (2012), the zonal

and meridional components of the Ekman transport are used to calculate $CSET$ from a discrete set of points parallel to the shoreline (Fig. 1, black dots):

$$CSET = -\sin(\phi)Q_x + \cos(\phi)Q_y \tag{3}$$

where $CSET$ is expressed in $m^2 s^{-1}$ and $\phi$ represents the angle between the shoreline and the equator. Whilst presenting a highly irregular topography, the coastline within CUS can be broadly approximated to $90°$ over IP, to $55°$ over MoUS and to

$90°$ off the SMUS coast relative to the equator (Alvarez et al., 2008 and Cropper et al., 2014). Positive (negative) values of $CSET$ correspond to upwelling-favorable (unfavorable) conditions.

The second mechanism contributing to upwelling is the Ekman pumping ($W_{ek}$) defined as:

$$W_{ek} = \frac{1}{\rho_w f}\nabla \times \tau \tag{4}$$

where $W_{ek}$ is expressed in $m^2 s^{-1}$, $\nabla \times \tau$ represents the curl of the derived wind stress vector integrated over the longitude

range $8°W$-$11°W$ off IP coast, $9°W$-$18°W$ along the Morocco coast and $16°W$-$20°W$ for the SMUS.

Finally, as highlighted in section 2.2, coastal upwelling may be modulated by the cross-shore geostrophic transport. We quantify this effect along the entire CUS as follows:

$$T_{geo} = MLD\frac{g}{f}(SSH_{north} - SSH_{south}) \tag{5}$$

where $T_{geo}$ is the vertical transport (in $Sv$) due to the zonal current generated from the meridional SSH gradient and $g$ is the gravity coefficient ($g$ = 9.81 m.$s^{-2}$). $T_{geo}$ is calculated right next to the coastal boundary, where it can interact with the vertical flux. Thus $SSH_{north}$ - $SSH_{south}$ is the difference between the northernmost and southernmost grid points close to the shore of the three different subregions of the CUS (see Fig. 1). In addition, the MLD is averaged between $37°N$-$43°N$ and $8°W$-$11°W$ off the IP, $21°N$-$25°N$ and $9°W$-$18°W$ for the nMoUS, $21°N$-$25°N$ for the sMoUS and over the same latitude range

of nMoUS and between $12°N$-$20°N$ and $16°W$-$20°W$ in the SMUS. Note that all indices described above are calculated over the native model grid. However, the metric used here (see section 2.4) to evaluate the models skill requires an interpolation. For this and only for the skill calculation (Fig.6) all the models have been interpolated from their native grids to a common





0.25°x0.25° lat-lon resolution grid, using a bilinear interpolation method. We noted that small changes are induced by the interpolation method (not shown) but this does not affect the skill scores in a statistically significant way.

## 2.4 Skill metrics

We use a metric to quantify the skill of the climate models at representing the CUS characteristics through the different upwelling indices. In this study we use the Arcsi-Mielke score (M ) previously used to evaluate the performance of High-ResMIP/PRIMAVERA model (Bador et al., 2020).

This is a non dimensional metric defined by:

$$M = (\frac{2}{\pi})\arcsin\left[1 - \frac{mse}{V_X + V_Y + (G_X - G_Y)^2}\right] \times 1000 \tag{6}$$

where $mse$ is the mean-square error, X and Y represent model and observed data respectively, $V$ is the spatial variance and $G$ the spatial mean. The Arcsin-Mielke score reaches a maximum possible value of 1000 when $mse$ is equal to 0, whereas a zero score indicates no skill and it can even be negative in the worst cases, although this rarely occurs. The skill score is computed separately over the different subdomains of the CUS and for the annual climatological averages of each upwelling index. To compute M all the upwelling indices have been interpolated on a common $0.25°x0.25°$ horizontal grid by using a bilinear interpolation method.

# 3 Characterization of the Canary upwelling system from observations and reanalysis

## 3.1 The thermal upwelling

The seasonal variability of the CUS upwelling intensity as described by the $UI^{sst}$ index is shown in Fig. 2 (left panels). Over the IP coast, the strongest positive values of $UI^{sst}$ are observed in summertime (July to September) and the index remains positive but weaker $(0.5°C)$ from November to June. This evolution is consistent with the available literature (Nykjær and Van Camp, 1994; Santos et al., 2005 and deCastro et al., 2008b). The ocean-coast gradient (i.e., the $UI^{sst}$ index) ranges from $1°C$ to $4°C$ from lat $21°N$ to $32°N$ (MoUS), with high values of $UI^{sst}$ through the whole year in sMoUS and during summer time (July to September) in nMoUS. The presence of the above mentioned high values of $UI^{sst}$ throughout the year in sMoUS are consistent with the permanent upwelling conditions described in previous works (Wooster et al., 1976; Barton et al., 1998 and Gómez-Gesteira et al., 2008). Further south, over the SMUS region, $UI^{sst}$ shows a marked seasonality with positive values of $UI^{sst}$ in winter (upwelling season) and negative values of $UI^{sst}$ in summer.

Fig. 2 reveals that although the three observational datasets present a similar behavior of $UI^{sst}$, substantial differences in the amplitude emerge. The largest discrepancies are found in sMoUS (whole year) and SMUS (winter) where $UI^{sst}$, values are significantly lower in ERSST-v5 than in the other datasets. In the rest of CUS subregions (i.e., IP and nMoUS), a stronger observational agreement is found.





### 3.1.1 Dynamical upwelling indices

Fig. 3 shows the seasonal cycle of $CSET$ for observations and reanalysis (left panels). Along the western coast of the IP, upwelling-favorable conditions (positive values of $CSET$) are observed during summer. This is coherent with the strengthen-

ing and northward displacement of the Azores high which promotes northerly winds. This marked upwelling season in summer is consistent with the results obtained from the thermal index (UISST) and with the previous research (Alvarez et al., 2005; Santos et al., 2005 and Gomez-Gesteira et al., 2006). As for $UI^{sst}$ (Fig. 2), negligible of even negative values of $CSET$ are detected along the IP coast during wintertime indicating the predominance of downwelling conditions.

Also consistent with previous studies (Gomez-Gesteira et al., 2006 and Benazzouz et al., 2014), $CSET$ is strong in summer in

nMoUS and permanent throughout the year in sMoUS. Finally, the SMUS is characterized by the existence of two well marked seasons: an upwelling season from approximately November to May and a downwelling season from June to October.

As for SST, the comparison between wind products also shows some discrepancies in terms of upwelling amplitude. Despite the CCMP wind dataset covers a shorter time period (1992-2011), there is no major difference with respect to ERA5. On the contrary, NOAA-20CR-v3 shows a slight enhance $CSET$ with respect to the other two datasets, particularly for SMUS and

nMoUS from April to June. A conclusion emerging from this analysis is that the Ekman transport might therefore depend on the underlying size of the grid cell. Thus, gridded datasets at different resolutions may lead to different estimates of the observed Ekman transport. This sensitivity of the Ekman transport to the spatial resolution is therefore crucial to properly compare modeled and observed upwellings.

Fig. 4 shows the seasonal cycle of the Ekman pumping ($W_{ek}$). Focusing on the validation datasets (left panels) and over the IP, $W_{ek}$ tends to be weak or practically null from October to June, and it is more intense (around 0. 5 $m^2 s^{-1}$) during summer season in CCMP and ERA5 as found by Alvarez et al. (2008). For NOAA-20CR-v3, this seasonal cycle is less marked. Along the MoUS, $W_{ek}$ is different in nMoUS and sMoUS sub-regions. In sMoUS, $W_{ek}$ is weak but positive with maximum values occurring during winter and spring. In nMoUS, validation datasets show mostly negative values of $W_{ek}$ throughout the year

as found by Lathuilière et al. (2008). This result may be linked by the fact that the meridional component of the wind stress ($V$) decreases (instead of increases) away from the coast. This results in negative $\frac{\partial V}{\partial x}$ in this region (not show) and favorable conditions for downwelling. Finally along the SMUS $W_{ek}$ is maximum in winter and spring. Therefore, the seasonal cycle of $W_{ek}$ is roughly that for $CSET$ (Fig. 3), with the main differences identified over the MoUS.

### 3.2 The onshore geostrophic flow and the quantitative assessment of the upwelling rate

As discussed in section 2, the effect of $T_{geo}$ is quantified here for the CUS. We have examined in the first time the SSH climatology from the AVISO satellite data and the GODAS reanalysis (see first two columns in Figure B1 of the appendix B). The SSH gradient observed over IP (panel a) is indeed negative all the year, thus potentially inducing an onshore geostrophic flow. The maximum and minimum amplitudes of this SSH gradient are found in summer and winter, respectively. Over MoUS (panels b and c), this gradient is negative all the year as for the IP and reach its maximum from May to November in the





nMoUS and from July to September (August to October) over sMoUS in AVISO (GODAS). In SMUS, the SSH difference is also always negative, but the related amplitude strongly differs among the distinct sub-regions analyzed (Fig. B1; panel d). The SSH difference is strong all year long in both datasets and tend to be maximum at the beginning of the upwelling season. Therefore, these SSH meridional gradients yield to an onshore geostrophic transport ($T_{geo}$) off the CUS during the upwelling season. It is important to mention that the latter term ($T_{geo}$) is counted negative eastward following the sign convention used to

quantify the upwelling (negative bars in Fig.5). The $T_{geo}$ is below 0.25 $Sv$ over IP (Fig. 5. a) in the three validation datasets. For nMoUS (Fig. 5. b) and sMoUS (Fig. 5. c) sub-regions, $T_{geo}$ is on average ∼0.25 $Sv$. Finally, the contribution of $T_{geo}$ is strongest over the SMUS, which presents values of 0.6 $Sv$ approximatively. This situation is mainly related to the fact that this sub-region shows stronger SSH gradients than the rest of the CUS (panel d Figure B1).

To estimate the total upwelling intensity ($UI_{total}$), we finally consider the integrated sum of the Ekman transport ($CSET$), the Ekman pumping ($W_{ek}$) and the geostrophic flow $T_{geo}$ (Jacox et al., 2018 and Sylla et al., 2019). Fig. 5.a (green bars) shows that, $UI_{total}$ over IP is positive and ranges between 0.25 $Sv$ and 0.45 $Sv$. $UI_{total}$ estimation leads to an total upwelling transport of 0.25 $Sv$ to 0.5 $Sv$ over nMoUS (Fig. 5. b) while it ranges from 0.5 $Sv$ to 1 $Sv$ over sMoUS (Fig. 5. c). Note that our previous analysis of Ekman pumping (Fig. 4) has shown negative values during the upwelling season (summer) in the nMoUS

favoring the predominance of downwelling conditions. The combination of both $W_{ek}$ and $T_{geo}$ may thus contribute to reduce the volume of upwelled waters due to the $CSET$ in this sub-region. In the SMUS (Fig. 5. d), where the Ekman divergence and the wind stress curl generate a significant vertical transport proceeds (Fig. 3 and Fig. 4), our estimation of $UI_{total}$ is about 1 $Sv$ to 1.5 $Sv$. This upwelling is however partially reduced (as in Sylla et al., 2019) by the strong effect of onshore transport.

## 4 Models evaluation

### 4.1 The thermal upwelling

To address the models analysis, we compare $UI^{sst}$ from the observational datasets (Fig. 2; left panels) and the different model configurations (Fig. 2; right panels). In the IP, there is a general agreement between observations and models. Models broadly reproduce the mean state of the mean seasonal cycle obtained in observations with a maximum in summer. Two exceptions can be noted: the CNRM-CM6 family shows no signature of upwelling with unrealistic negative values of $UI^{sst}$ during summer,

and the CMCC-CM2 family for which the seasonal cycle can barely be identified in its both versions. In general, the amplitude of the seasonal cycle is enhanced when both the ocean and atmosphere resolution are increased (comparison among groups 1 and 2).

Along the nMoUS and sMoUS, the group 2 provides a more realistic representation of this SST index than their LR versions (group 1). In the latter case the amplitude is markedly underestimated over the sMoUS sub-region and the maximum of $UI^{sst}$

observed in summer is not reproduced in nMoUS except for the CNRM-CM6-1-LR model. For both groups 1* and 2* however, the upwelling is not reproduced in nMoUS, while in sMoUS these groups broadly reproduce the permanent upwelling with an overestimation of $UI^{sst}$ amplitude in MPI-ESM1. Thus, the only increase of the atmospheric resolution in models produces





no clear impact on upwelling representation.

Along the SMUS sub-region and for the group 1 (i.e., the LR model versions), the upwelling season seems to be longer

that the observed: it starts earlier (October) and ends later (June), with a marked drop for CMCC-CM2 (group 1*). However both, MPI-ESM1-2-HR and MPI-ESM1-2-XR, simulate a realistic seasonal cycle in comparison with the observations, but the corresponding amplitudes are largely overestimated over the sMoUS and the SMUS. This situation may can be explain by the results found in Gutjahr et al. (2019). According to these authors the MPI model suffers a severe cold bias in the whole northern hemisphere and, particularly, in the Atlantic sector. In this line, Roberts et al. (2019) show that a higher-resolution atmosphere

tends to produce a cooler ocean SST, particularly in the ocean upwelling regions (in agreement with Gent et al., 2010 and Small et al., 2014). This cooling has been already assessed by Putrasahan et al. (2019) and is caused by a slowed AMOC due to the underestimation of the wind stress, the northward heat, and the salt transport. The above mentioned deficiencies in the representation of the seasonal cycle seem to be improved when both the atmospheric and the ocean resolution increase (group 2). On the contrary, when just the atmospheric resolution is increased they persist (group 2*).


In summary, increasing the horizontal resolution of the atmosphere or both, the atmosphere and the ocean, alters the representation of the CUS if it is characterized with the thermal index $UI^{sst}$. Nevertheless, different features are identified along the distinct sub-regions within the CUS. Thus, the IP does not seem to be very sensitive to these changes in model resolution. On the contrary, upwelling representation in the western African coast (MoUS and SMUS) improves when ocean and atmospheric

resolution enhance. This is consistent with Ma et al. (2019) and Balaguru et al. (2021) who found that an increase in horizontal resolution can potentially reduce the warm bias of climate models along these regions through an improved simulation of coastal upwelling. This responds to the fact that thermocline rises more sharply near the coast causing a reduction of the near-shore SST bias in the high-resolution global climate models. On the other hand, the comparison of group 1* and group 2* indicates that the upwelling estimated with $UI^{sst}$ does not change significantly when only the atmospheric resolution is

increased. Therefore, we infer that enhancing ocean resolution is required to improve the SST-related upwelling index representation over the CUS. This is in agreement with previous studies such as Small et al., 2015 for the Benguela upwelling system. Additionally, Gutjahr et al. (2019) suggest that a high spatial resolution in the ocean reduces the bias in both, the ocean interior and the atmosphere. All this leads to the important conclusion that a high-resolution ocean plays a key role for properly representing the ocean and atmosphere mean states.

## 4.2 Dynamical upwelling indices

As for the thermal index $UI^{sst}$, we evaluate the ability of the different model configurations to reproduce the seasonal variability of $CSET$ (Fig. 3) and $W_{ek}$ (Fig. 4). Along the IP coast all model configurations show the seasonal of $CSET$ with the maximum during summer. However in group 1, the upwelling period is in general overestimated, which is not the case of group 2. Regarding groups 1* and 2*, no major difference has been identified, which therefore difficult to extract a relationship

with confidence.

Focusing on the sMoUS and nMoUS, group 1 largely overestimates $CSET$ in these subdomains, being this overestimation less





well established for group 1* (Fig. 3). On the contrary, group 2 is broadly coherent with the validation datasets and the group 2 * also provides more realistic $CSET$ values than group 1*. This suggest that higher resolution winds lead to an slightly improved Ekman transport. Similar conclusions are generally drawn over the SMUS region for groups 1* and 2* whereas the

group 2 shows a better agreement with the validation datasets than group 1. The latter again overestimates the amplitude of the Ekman transport with respect to the reference values. This situation has also been documented in Castaño Tierno (2020).

Let's now consider the ability of the different model configurations to reproduce the seasonal variability of the wind stress curl (Fig. 4). The models reveal sometimes noisy patterns, making complex in these cases the interpretation of the effect of

model resolution, but some conclusions can be drawn. Group 2 reproduces the expected larger features of the $W_{ek}$ seasonal cycle in the different sub-regions, and a comparison with group 1 reveals differences in structure and intensity particularly over the southern flank (MoUS and SMUS). Group 1 shows generally a rather sharp and unrealistic seasonal cycle in SMUS which is longer than that identified in the validation datasets. The improvement in group 2 may be linked to that found by Ma et al. (2019): a finer horizontal resolution of climate models enables better representation of low-level coastal jet structure, with

stronger and closer alongshore wind stress and curl leading to a more realistic representation of upwelling. The refinement of just the atmospheric resolution (group 2*) also leads to an improved wind stress curl.

### 4.3    Geostrophic flow and total upwelling transport

Fig. 5 shows the total upwelling transport ($UI_{total}$) by taking into account the different upwelling terms: $CSET$, $W_{ek}$ and $T_{geo}$. The latter (computed as described in section 2. 3 and represented in Fig. 5 with negative bars) is too weak in group 1

over the IP coast, which is related to the low contribution of SSH gradient in these models during the upwelling season (Fig B1 in Appendix B). When both the ocean and atmosphere resolution are increased (group 2), the realism of $T_{geo}$ is improved. It is broadly consistent with the observational estimates. The CMCC-CM2 family in group 1* and 2* provides realistic $T_{geo}$ values independently of the model resolution. However, this onshore transport is very low (close to zero) in MPI-ESM1 models for both resolutions particularly due to the shallower mixed layer depth (not shown) over the North Atlantic. This feature

is consistent with Gutjahr et al. (2019). The effect of increasing only the atmospheric resolution is, therefore, difficult to be established. As in group 1 this is probably due to the relatively weak effect associated with the SSH-related contribution. In nMoUS and sMoUS (Fig. 5 panels b and c), the role of the resolution on the simulated $T_{geo}$ is not clear and the difference amongst the groups is very small. Finally in the SMUS region, the more realistic estimates of $T_{geo}$ are generally provided by the group 2 as also the simulated SSH gradient (Fig B1). For groups 1* and 2*, the MPI-ESM1 models show better agreement

with observations than the CMCC-CM2 models, but the impact of the atmospheric resolution is again not conclusive.

Let's now consider the total upwelling transport ($UI_{total}$) computed as the sum of all dynamical effects, as explained in section 3.2. We find, along the IP coast (Fig. 5, panel a), that both groups 1 and 1* markedly overestimate the upwelling total transport. This overestimation is reduced in group 2 with $UI_{total}$ values slightly higher than the observational range (horizontal

dashed lines). However minor differences are found among groups 1* and 2 *.





In the nMoUs, the group 1 and group 1* again largely overestimate the total upwelling transport except for the CNRM-CM6-1-LR which shows values almost close to zero and CMCC-CM2-HR4 which provides a generally good estimation of $UI_{total}$. The very weak value in CNRM-CM6-1-LR can be explained by the downwelling effect displayed by the $W_{ek}$, which is relatively strong in this model configuration (Fig. 4). On average, the HR model versions (group 2) perform better the $UI_{total}$, which appears within the range of observational estimates. Models of group 2 are generally in the range of observational estimates and the group 2* shows a small improvement with increased resolution. The CMCC-CM2-VHR4 is close to the validation datasets as its LR version, while MPI-ESM1 overestimate always $UI_{total}$, although the value is smaller in the HR version than in the LR version.

In the sMoUS sub-region the differences among models resolution are less marked than in the previous regions. Thus, it is difficult to directly relate the representation of $UI_{total}$ with model resolution. Finally over the SMUS domain and as it is seen in Fig. 5 panel d, the group 2 have a general better agreement with the observations than the group 1, for which the difference with observations remains clear and outside the range of the observed $UI_{total}$. This means that these models group are able to fully capture the estimation of the upwelling transport by realistically representing all the dynamical indices in such a way that the simulated upwelling in this sub-region tend to be systematically more tightly clustered. Our results for groups 1* and 2* goes in the same direction with a similar range of $UI_{total}$ and no clear effects are identified.

### 4.4   Quantitative measure of the model skill

In this section, we evaluate quantitatively the performance of the models in simulating the CUS using the Arcsin-Mielke M score (see section 2. 4). M is computed between the observed and the simulated upwelling indices as a function of the nominal ocean resolution. The reference datasets are OISST-v2 for the thermal index (SST-based) and ERA5 for the dynamical (wind-based) indices. Note that the skill score values may be sensitive to the choice of the observational datasets to measure the model performance in some cases (Figure C1 in Appendix C). For instance, for most of sub-domains and indices, differences in skill scores computed with HadISST or OISST-v2 (ERA5 for dynamical indices) for $UI^{sst}$ are as large as 250 points between the M skill scores computed with ERSST-v5 (NOAA-20CR-v3), but the slopes of the lines that connect the skill scores at different resolutions remain unchanged. Observational consistency is quantitatively assessed using the average of the M scores computed from each possible combination of pairs of observational datasets. This consistency is represented by the horizontal lines on panels in Fig. 6. It is moderately high (above 300 points) for the thermal index (top panels) and Ekman transport (central panels) in most of the sub-regions analyzed. However this value is very low for the Ekman pumping (Fig. 6; bottom panels) in the case of IP and nMoUS. This feature indicates, for these sub-domains, a weaker similarity among the validation datasets. These results illustrate the challenge that exists in providing an accurate characterization of the upwelling systems in observations and reanalysis.

For $UI^{sst}$ (top panels), the slopes of the lines that connect the group 1 and group 2 in the MoUS and SMUS (panels b, c and d) sub-regions are negative, indicating a higher skill for higher resolution (group 2). However, an opposite behavior is





observed over the IP (panel a), where low resolution models present larger levels of skill. We note no robust change for the M scores between group 1* and group 2*. These results support the conclusions drawn from Fig. 2 in section 4.1. Let's try now to decompose the total Ekman process: for the Ekman transport (Fig. 6; central panels), again the slopes of the lines that connect M values indicate a higher skill for group 2 in the MoUS and SMUS. For group 1* and group 2* models results's are still limited in reaching systematic conclusions on the effect of enhanced resolution in the atmospheric component although

the MPI-ESM1-2-HR clearly shows larger skill score than its LR version in the nMoUS and SMUS sub-regions. Along the IP sub-region, the M scores are not conclusive, except for the CNRM-CM61-HR which provides a higher score with increased resolution.

Results regarding the Ekman pumping (Fig. 6; bottom panels) are similar to those obtained for the Ekman transport, with improved model performance as the both resolutions increases and no systematic response when only the atmospheric grid

resolution is modified. However, M values are broadly lower for the Ekman pumping than for the Ekman transport, indicating that models are less efficient in capturing the Ekman pumping than the transport along the coast. The conclusions obtained from the M scores corroborate the results found in the previous section for the individual upwelling indices.

To summarize, the group 2 provides a higher skill score than the group 1 in the Morocco and SMUS upwelling systems, but no significant improvement in simulating the Iberian Peninsula upwelling system is found. On the other hand, increasing atmo-

sphere resolution (group 2*) has a limited effect on the skill scores. Nevertheless, it is necessary to note that only two models form this group 2*, which make it difficult to extract a significant relationship between increased atmosphere resolution and model performance.

## 5    Summary and Conclusions

Climate models provide, by construction, imperfect representation of the climate system. In particular, and because of their coarse resolution, the performance of global climate models to simulate the coastal upwelling systems is subject of a wide discussion. The biases of the climate models with both an oceanic and/or atmospheric origin are closely linked to limitations in the model physic formulation and insufficient model resolution (Li and Xie, 2012; Zuidema et al., 2016 and Harlaß et al., 2018b). Nevertheless, and are useful to assess, it has been shown that coupled models (CMIP5/CMIP6) are able to reproduce

some features of these systems and are used to assess changes in the future (Wang et al., 2015 and Sylla et al., 2019). The upwelling phenomenon is one of the physical processes most sensitive to the model resolution and for which an improvement is expected when the resolution is increased (Small et al., 2015 and Vazquez et al., 2019). This study provides the first attempt to systematically evaluate the effect of increasing global model resolution (both in the atmosphere and ocean components) on the representation of the CUS. We have analyzed the historical simulations from six global climate models following the High-

ResMIP protocol (Haarsma et al., 2016). Four upwelling indices based on SSTs, wind stress and sea surface height have been used as metrics to assess the effects of increased models resolution. A quantitative skill metric, the Arcsin mean skill score, has also been applied to measure the models performance with respect to observational datasets. The most relevant findings can be



summarised as follows:

Globally, our results show that observations and reanalyses yielding a fairly consistent picture of the CUS climatology, regardless of their resolution. However in the southern part of Morocco and in the Senegalo-Mauritanian areas, upwelling indices derived from the validation datasets at lower resolution (ERSST-v5 and NOAA-20CR-v3) show greater magnitudes than those derived from the higher resolution datasets. The average of the M skill scores used to quantify the consistency among observational and reanalysis datasets at different resolutions is not very high. This highlights the challenge that exists for choosing
a proper observational dataset to evaluate global climate models performance.

The impact of increasing model resolution is not the same in the different sub-domains of the CUS. In the northern part, within the IP domain, the high-resolution models do not seem to better simulate the structure of the climatological SSTs and the winds linked to the upwelling. For some models the LR version even produces better results than the HR version. However,
in the southern CUS, and in particular in the MoUS and SMUS, the HR models show a clear improvement in the representation of upwelling indices. Increasing the resolution leads to simulations that are in better agreement with the observations. The best results are obtained when the resolution is increased for both components of the couple models, ocean and atmosphere. According to our results, the effect of increasing only the atmospheric resolution is not clear. This is probably mainly due to the fact that the sample analyzed in this case is small (only two models). The results presented here suggest nevertheless that
increasing the resolution of the atmospheric component is not enough, and that it is also necessary to increase the resolution of the ocean to obtain a significant improvement in the representation of the CUS. The oceanic resolution emerges therefore as a key factor for having more realistic simulations, which is in agreement with other studies (e.g. Bryan et al., 2010; Putrasahan et al., 2013; Parfitt et al., 2017 and Bellucci et al., 2021). Our results are also in line with previous modeling work suggesting that an increased resolution improve global climate model performance. Roberts et al. (2019) have shown that, increased model
resolution in the atmosphere and ocean can have considerable impact on the large tropical Atlantic biases seen in typical CMIP-resolution models of the mean state and variability, both at the surface in terms of temperature, as well as in the deeper ocean. According to Czaja et al. (2019), a clear dependence on resolution (both ocean and atmosphere) is found and better agreement with reanalysis and observations. However, this issue may also be model and region dependent (Delworth et al., 2012 and Raj et al., 2019). In the present study, the representation of oceanic processes related to upwelling has not been investigated in
detail. Further work is needed to better understand the role of the ocean dynamic on the simulated upwelling improvements, in particular for comparing groups 1 and 2. The study of ocean stratification and vertical transport can provide insight into the relative role of increased ocean and atmospheric resolution in improving the representation of upwelling.

This study provides encouraging results for high-resolution global climate modelling, although many aspects related to the
physical processes must be further assessed in the future. However, as already argued in previous studies that have analyzed HighResMIP simulations (Bador et al., 2020; Moreno-Chamarro et al., 2022 and López-Parages and Terray, 2022), increasing the resolution of a global climate model does not necessarily have to be the only way to better represent the climate system.



There is still much work to be done in terms of physical parameterizations as suggested by Patricola et al. (2012) and Harlaß et al. (2018a). The improvements in model parametrizations and process representations, specific corrections applied to models,

additional tuning, and longer spin-ups might all be essential. On the other hand, climate variability is particularly important in the near term, and for highly variable quantities such as precipitation. But this might not be the case of coastal upwelling. Indeed, individual members of high resolution model show no difference to simulate the Canary upwelling system (not shown). We might infer that individual model runs do not necessarily represent independent estimates and therefore, it may be more convenient to only run a small subset of ensemble members for models at high resolution (although computationally expensive)

than a large subset of ensemble members for models at standard resolution. This and other related question must be necessarily faced in future works.

*Code and data availability.* All of the HighResMIP/PRIMAVERA model output analyzed in this manuscript is openly available from the Earth System Grid Federation (ESGF; https://esgf-index1.ceda.ac.uk/search/cmip6-ceda/) via the references. The SSTs products used in this analysis are available at https://psl.noaa.gov. The wind products are available via https://podaac.jpl.nasa.gov and https://apps.ecmwf.int/datasets/data/interim-

full-daily. The SSH datasets used in this manuscript are currently available at https://www.esrl.noaa.gov/psd/gridd ed/data.godas.html and https://www.aviso.altimetry.fr/en/data.html.

*Author contributions.* AS is the first author of the manuscript and performed the largest part of the analysis. ESG, JM and JLP contributed with the ideas and writing and approved the submitted version.

*Competing interests.* The authors declare that there is no conflict of interests regarding the publication of this paper.

*Acknowledgements.* This work is carried out as part of TRIATLAS project (South and Tropical Atlantic climate-based marine ecosystem prediction for sustainable management) funded by the European Union's Horizon 2020 research and innovation programme under grant agreement No 817578.





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

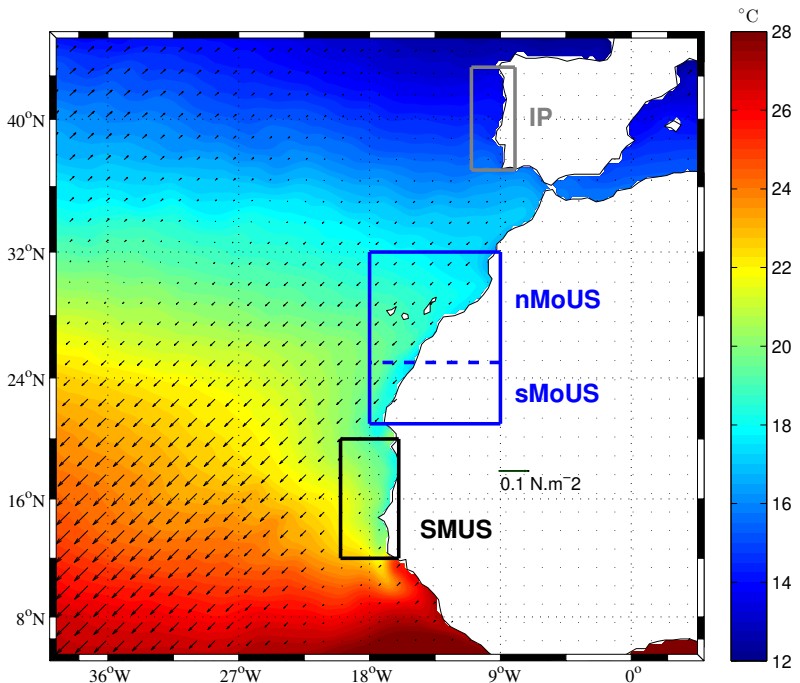

**Figure 1.** Colors: OISSTv2 climatological mean ($^\circ C$) in March averaged over 1992-2011 period. Black vectors show the wind stress from CCMP (computed offline from winds as described in Appendix). The referent vector is shown in land. The grey box (37°N-43°N and 8°W-11°W) represents the Iberian Peninsula region (IP); the blue box (21°N-32°N and 9°W-18°W): the Morocco (MoUS) area with a dashed line separating nMoUS and sMoUS. The black box (12°N-20°N and 16°W-20°W) represents the Senegalo-Mauritanian sub-region (SMUS). The stars indicate the coastal (black) and offshore (magenta) locations used for the computation of thermal upwelling index (see section 2. 3). The black and magenta dots are separated by 5° of longitude.



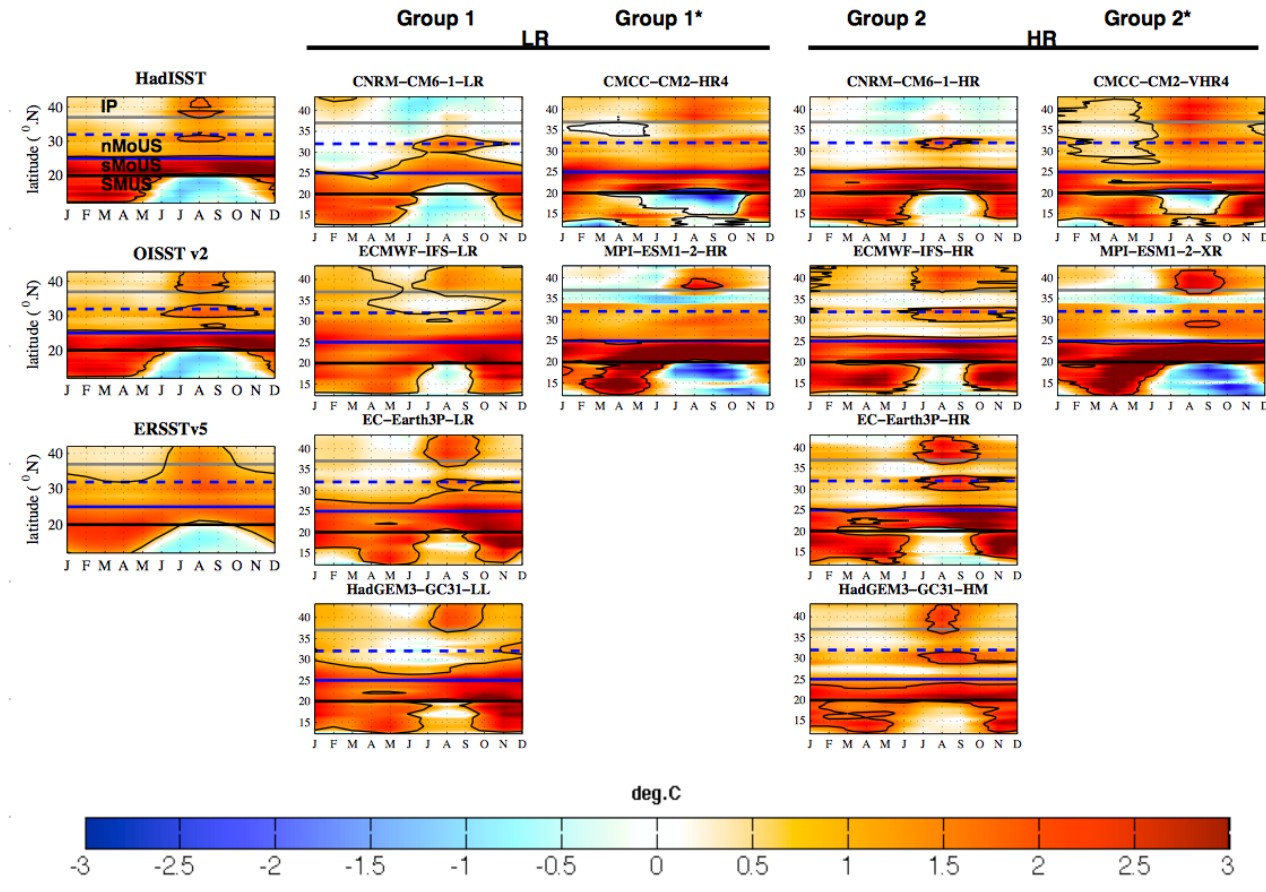

**Figure 2.** Seasonal cycle of $UI^{sst}$ upwelling index ($°C$) calculated as explained in section 2.3 for the period 1985-2014 and shown here as a function of the latitude for several models configurations and reference datasets (HadISST, OISST-v2 and ERSST-v5). Following columns: models from the groups 1, 1*, 2 and 2* respectively (see section 2.1 for the definition of these groups). This index is calculated over the models native grid.



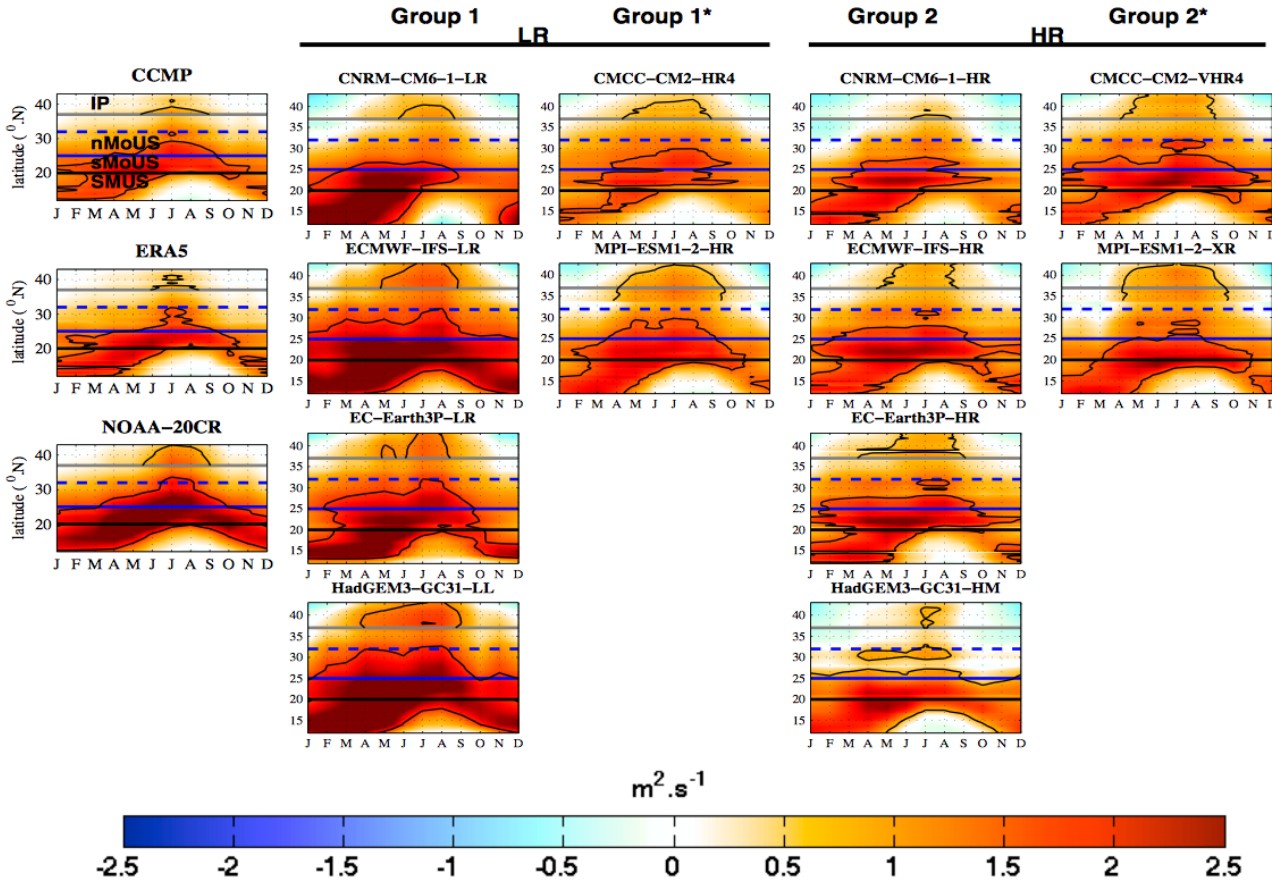

**Figure 3.** Seasonal cycle of $CSET$ upwelling index ($m^2 s^{-1}$) calculated as explained in section 2. 3 for the period 1985-2014 and shown here as a function of the latitude for several models configurations and reference datasets: ERA5, NOAA-20CR-v3 and CCMP (for the period 1992-2011). Following columns: models from the groups 1, 1*, 2 and 2* respectively (see section 2.1 for the definition of these groups). This index is calculated over the models native grid.



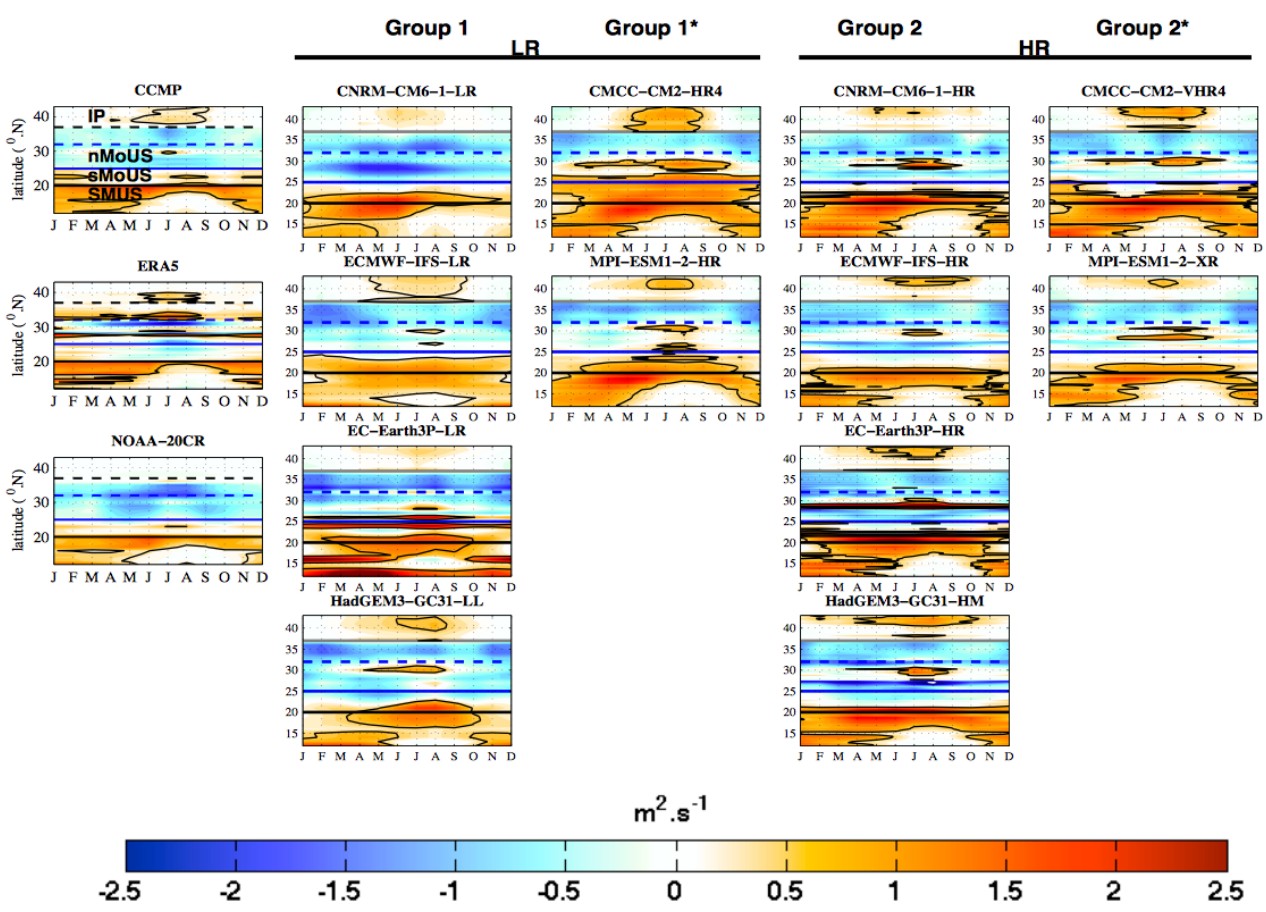

**Figure 4.** Seasonal cycle of $W_{ek}$ upwelling index ($m^2 s^{-1}$) calculated as explained in section 2. 3 for the period 1985-2014 and shown here as a function of the latitude for several models configurations and reference datasets: ERA5, NOAA-20CR-v3 and CCMP (for the period 1992-2011). Following columns: models from the groups 1, 1*, 2 and 2* respectively (see section 2.1 for the definition of these groups). This index is calculated over the models native grid.



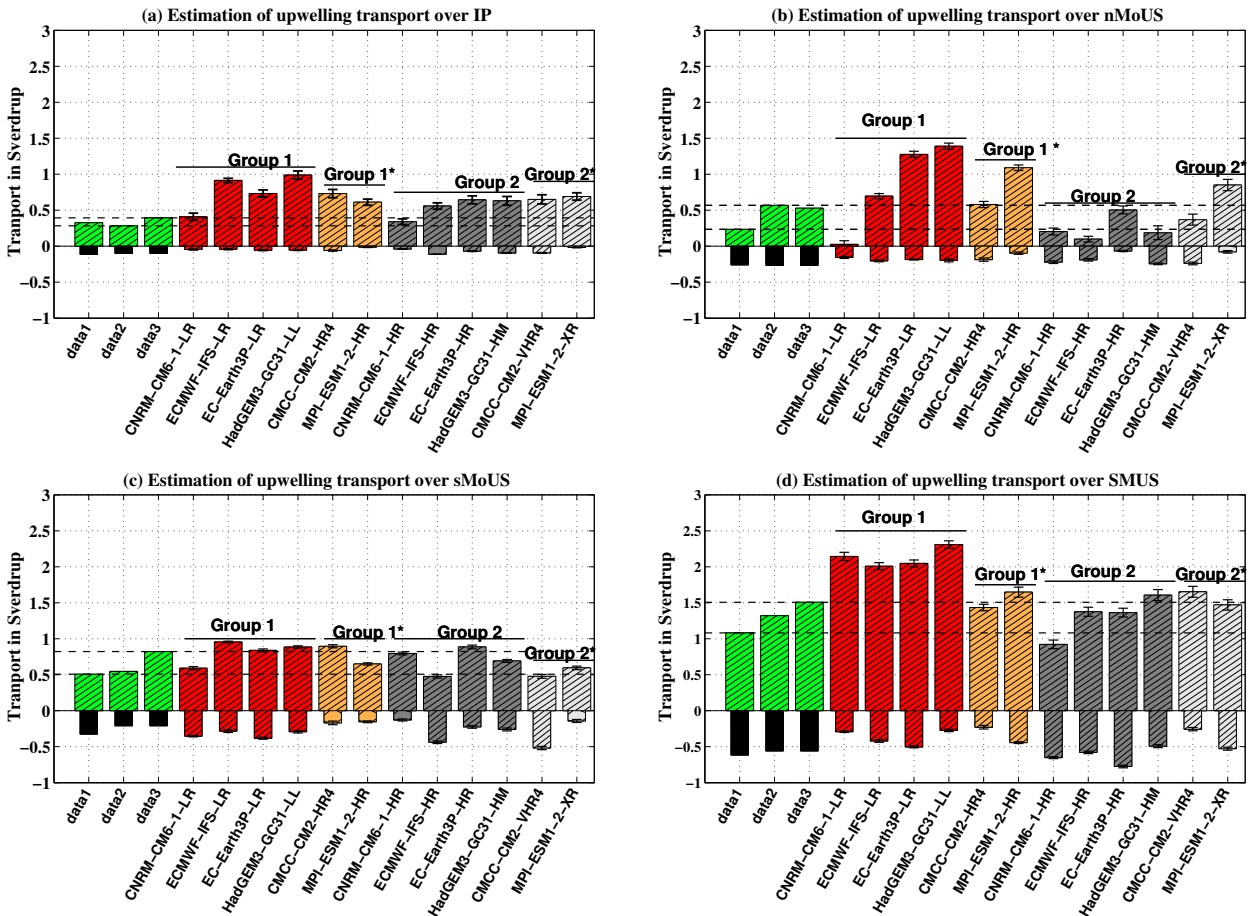

**Figure 5.** Negative bars: estimate of the seasonal integrated contribution in Sverdrup (±error bars: black whiskers bars) of $T_{geo}$ computed from Eq (7) and averaged from July to September along the IP (37°N-12°N and 8°W-11°W, panel a) and in nMoUS (26°N-32°N and 9°W-18°W, panel b), panel c: all year in sMoUS (21°N-25°N and 9°W-18°W) and from November to May along the SMUS (12°N-20°N and 16°W-20°W, panel d). The first black bar shows $T_{geo}$ computed from AVISO satellite data period (1993-2005) and mld (de Boyer Montégut, 2004), and the second and third black bars correspond to $T_{geo}$ derived from the GODAS reanalysis (1985-2014) and the previous MLD data. The following columns show the results for the individual climate models. Positive bars display the total volume of upwelling water ($UI_{total}$) computed as the sum of the three contributions of the three dynamical indices ($CSET + W_{ek} + T_{geo}$) to the upwelling. Data 1 (1993-2005) corresponds to Ekman process computed from CCMP and $T_{geo}$ from AVISO SSH product and MLD de Boyer Montegut. Data 2 and data 3 represent the Ekman process from ERA5 and NOAA-20CR-v3 over 1985-2014 respectively and $T_{geo}$ from GODAS and the same MLD used in data 1. The horizontal discontinuous lines highlighted the observational range.



**Figure 6.** M skill score of SST gradient between ocean minus coast (top panels), Ekman transport (central panels) and Ekman pumping (bottom panels) as a function of the ocean model nominal resolution. The models from the group 1* and 2 * are thus represented by the vertical lines (with their HR version: MPI-ESM1-2-XR and CMCC-CM2-VHR4 highlighted by the light blue and magenta color respectively). The score is computed and averaged along the IP sub-region (panel a) and nMoUS (panel b) from July to September, all the year in sMoUS (panel c) and from November to May along the SMUS (panel d) over the period 1985-2014. For the SST index (dynamical indices), each model is evaluated to OISST-v2 (ERA5) observation. Horizontal lines on the each panel correspond to the average of the M scores computed from all combinations of pair of observational products.



| | Model name | Atmosphere nominal resolution | Ocean nominal resolution |
|---|---|---|---|
| **Group 1 (LR-ocea/atm)** | CNRM-CM6-1-LR | 2.5° | 1° |
| | ECMWF-IFS-LR | 0.5° | 1° |
| | EC-Earth3P-LR | 1° | 1° |
| | HadGEM3- GC31-LL | 2.5° | 1° |
| **Group 1* (LR-atm)** | CMCC-CM2-HR4 | 1° | 0.25°. |
| | MPI-ESM-1-2-HR | 1° | 0.4°. |
| **Group 2 (HR-ocea/atm)** | CNRM-CM6-1-HR | 0.5° | 0.25° |
| | ECMWF-IFS-HR | 0.25° | 0.25° |
| | EC-Earth3P-HR | 0.5° | 0.25° |
| | HadGEM3- GC31-HM | 0.5° | 0.25° |
| **Group 2* (HR-atm)** | CMCC-CM2-VHR4 | 0.25° | 0.25° |
| | MPI-ESM-1-2-XR | 0.5° | 0.4°. |
| **Variables used**: tos, tauuo, tauvo, zos, mlotst | | | |

**Table 1.** List of the HighResMIP/PRIMAVERA models used in this study. The first and second columns list the groups and models used. The third and fourth columns indicate the atmosphere and ocean nominal resolution. The last row at the bottom of the table lists the variables that were used for our study: sea surface temperature (sst, called tos in HighResMIP database), zonal and meridional wind stress components (tauuo and tauvo), sea surface height (ssh, called zos), mixed layer depth (mld, called mlotst).



| Variables used | Name of datasets | Period | spatial reoslution |
|---|---|---|---|
| SST (observations) | HadISST.2 (Titchner et al.2014) | 1981-2016 | 0.25° x 0.25° |
| | OISSTv2 (Reynolds et 2007) | 1982-2015 | 0.25° x 0.25° |
| | ERSST v5 (Huang e al, 2017) | 1854-2019 | 2° x 2° |
| Surface wind | Quikscat (Freilich et al., 1994; http://podaac.jpl.nasa.gov database) | 2000-2009 | 0.25° x 0.25° |
| | CCMP satellite data (Atlas et al., 2011) | 1992-2011 | 0.25° x 0.25° |
| | ERA5 reanalyse (Hersbach et al., 2020) | 1979-2019 | 0.25° x 0.25°. |
| | NOAA-20CR v3 reanalyse (Slivinski et al.., 2019) | 1836-2015 | 1° x 1° |
| Sea surface height | AVISO satellite data (Ducet et al., 2000; www.aviso.altimetry.fr) | 1995-2005 | 0.25° x 0.25°. |
| | GODAS reanalyses (https://www.esrl.noaa.gov /psd/gridded/data.godas.ht ml) | 1980-2020 | 1° x 1° |
| Mixed layer depth | de Boyer Montegut 2004 | climatology | 2° x 2°. |

**Table 2.** Observations and reanalysis datasets selected for this study and specifications of their resolution and coverage period.



## Appendix A: Comparison of wind products and offline estimation of stress wind

We compare the wind datasets used in this study by performing a preliminary analysis of the wind roses over the CUS region (Fig. A1). To simplify the datasets comparison, we consider the months (August for the IP and MoUS and February for SMUS) for the when upwelling occurs in these sub-regions. Along the Iberian Peninsula coast (first column) and over the Morocco (second column) to the Senegalese-Mauritanian coast (last column) the trade winds blow from the north-west to north-east approximately 10% to 35% and 15% to 60% of the select time at speeds ranging between 2.5 to 5 m/s and between 5 to 10 m/s respectively. We note a good similarity among wind datasets, across all the considered domains. Quikscat slightly overestimates the wind speed over the SMUS. Therefore, we chose to work with CCMP because it covers a larger period of time than Quikscat. Additionally the agreement between ERA5 and NOAA-20CR-v3 with the observations provide good support for using these reanalysis, exactly matching the present period considered in the climate models (1985-2014).

For all validations data, the wind stress was computed using the bulk formula as Santos et al. (2012) and Sylla et al. (2019):

$$\tau_x = \rho_a C_d (uas^2 + vas^2)^{1/2})uas \quad and \quad \tau_y = \rho_a C_d (uas^2 + vas^2)^{1/2})vas \qquad (A1)$$

where $uas$ and $vas$ are the zonal and meridional wind components respectively, $C_d$ the drag coefficient ($C_d$=0.0014 and $\rho_a$ the air density ($\rho_a = 1.22 kgm^{-3}$).



**Figure A1.** Wind rose diagram in August and averaged over the period [2000-2009] along the Iberian Peninsula (first column) and Morocco (second column) sub-regions and in February along the Senegalo-Mauritanian (last column) from Quikscat, CCMP, ERA5 and NOAA-20CR-v3 wind products. The concentric circles represent a different frequency range, ranging from zero at the center to increasing frequencies at the outer circles. The different colors provide details on the wind speed (in m/s) for each direction.



## Appendix B:  A counteracting effect: contribution of the Sea Surface Height

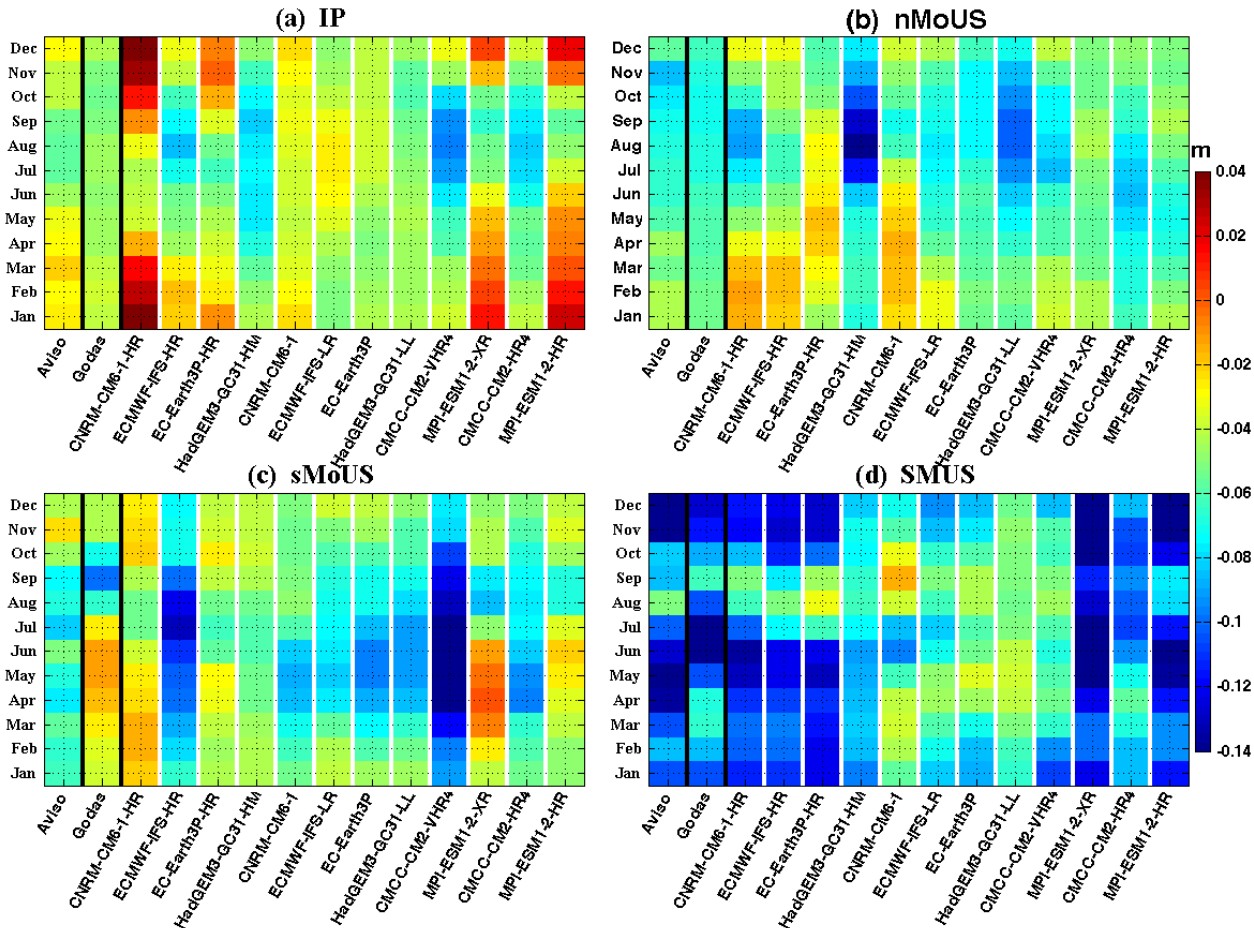

**Figure B1.** Monthly climatology of the meridional sea surface height difference (units: $m$) between coastal SSH values at the northernmost and southernmost grid points close to the shore over the Iberian Peninsula (panel a), north and south Morocco sub-domains (panels b and c respectively) and in the Senegalo-Mauritanian sub-region (panel d). The first two columns on the left (highlighted in black) show respectively, the results from AVISO satellite data [1993-2005] and GODAS reanalysis [1985-2014]. The other bands show the individual HighResMIP models.



765 **Appendix C:  Comparaison of model skill for different reference datasets**



**Figure C1.** M skill score of upwelling indices as Fig. 6 with each model evaluated to the SSTs datasets (OISST-v2, HadISST and ERSST-v5) for the thermal index and for the dynamical indices to ERA5 and NOAA-20CR-v3.