# Peer review of "Impact of increased resolution on the representation of the Canary upwelling system in climate models"

_Geoscientific Model Development, 2022_

## Author Comment (AC2)

[Figure]

Figure 1: Estimation of upwelling transport (sverdrup) averaged from November to May over 1985-2005 period computed by the integration of Ekman transport along the boundaries of SMUS region (orange, 12°N-20°N and 16°W-20°W) and estimation computed with the sum of Ekaman divergence and Ekman suction (cyan). The left (right) column in black (magenta) correspond to the validation data (Multi-model mean: MMM).

[Figure]

Figure 2: Amplitude of SST seasonal cycle along the Canary upwelling system and averaged over the period 1985-2014 in the Highresmip models and OISST v2 observation.

---

## Author Comment (AC3)

[Figure]

Figure 1: Estimation of upwelling transport (sverdrup) averaged from November to May over 1985-2005 period computed by the integration of Ekman transport along the boundaries of SMUS region (orange, 12°N-20°N and 16°W-20°W) and estimation computed with the sum of Ekaman divergence and Ekman suction (cyan). The left (right) column in black (magenta) correspond to the validation data (Multi-model mean: MMM).

[Figure]

Figure 2: Monthly climatology of the Ekman pumping $(m^2 s^{-1})$ for the period 1985-2014 and shown here as a function of the latitude for several models configurations and reference datasets: ERA5, NOAA-20CR-v3 and CCMP (for the period 1992-2011). Following columns: models from the groups 1, 1*, 2 and 2* respectively. In each panel the black contour shows the contour 0.75 and 0.5. The Ekman pumping over Morocco is here computed in the box defined in Fig. 1 (new version)

---

## Author Response (AR2)

**Dear editor and reviewers**

We thank you for your comments on our submitted manuscript. We answer below each of the points raised by the reviewers and we have put effort in generally improving the language and phrasing of the manuscript. **Our answers appear in blue**. We also submitted a fully revised manuscript. We hope that you will be convinced. Thanks again for your efforts Adama SYLLA and co-authors

Anonymous Referee #1: Review of Manuscript gmd-2022-130

Title: Impact of increased resolution on the representation of the Canary upwelling system in climate models.

Authors: A. Sylla, E.S. Gomez, J. Mignot and J.L. Parages

Recommendation: major revision

Summary

This study tested the state-of-the-art CMIP-class Earth system models (ESMs) to what extent the models can reproduce the Canary upwelling system along the coast from the Iberian Peninsula to the northwestern Africa, which is one of the areas of marine ecosystem and fishery. In general, ESMs with coarse resolution (1-2 degrees) fail to reproduce the coastal upwelling system due to several causes like wind stress, its curl, heat fluxes, etc. The authors also analysed Highres-MIP data (- 0.25 degree) and showed the benefits of refinements of atmospheric and oceanic horizontal resolution being consist with previous studies that focus on other upwelling areas. Interestingly, the authors employ several metrics to describe the coastal upwelling quantitatively and the results based on this methodology are well summarized. Therefore, I would think that this study would have feedbacks on model development and insightful understandings on the coastal upwelling in model simulation. On the other hand, I have several (most of them should be minor) concerns about plottings and interpretations on the results. As below, I am providing my comments and would expect the authors to address them and revise the manuscript. After adequate revision, this manuscript will be accepted as a publication in GDM.

We thank the reviewer for his general comment and appreciation of our manuscript. We answer below each of his/her points.

Minor comments:

1) Line 2. operating => operated?

Thanks, the word "operating" was replaced by "operated" in the new manuscript.

2) Lines 3-4. Might delete For this project the resolution of the ocean/or atmosphere components was increased.

Sentence was removed in the new manuscript, thanks

3) Lines 19-20. Some references should be added.

Thanks, references were added in the new manuscript:

Herbland, A., Voituriez, B., 1974. La production primaire dans l'upwelling maur- itanien en mars 1973. Cah. O.R.ST.OM., Sér. Océanogr. 12 (3), 187–201.

Minas, H.J., Codispoti, L.A., Dugdale, R.C., 1982. Nutrients and primary production in the upwelling region off Northwest Africa. Rapp. P.-V. Reun., Cons. Int. Explor. Mer 180, 148–183.

Tretkoff, E. (2011). Research Spotlight: Coastal cooling and marine productivity increasing off Peru. Eos Transact. Am. Geophys. Union 92, 184–184. doi: 10.1029/2011eo210009

Huyer, A. (1983). Coastal upwelling in the California Current system. Prog. Oceanogr. 12, 259–284. doi: 10.1016/0079-6611(83)90010-1

4) Line 27. "induces a positive wind stress" talking about only NH? If SH is included, better to say "cyclonic" wind stress curl.

The word "positive" was removed and replaced by "cyclonic" in the new manuscript

5) Line 36-37. Synoptic. For me, "synoptic" sounds more spatial. But, maybe the authors want to mention temporal variability here, I suppose.

Thanks, for this remark we have reformulated the sentence in the new version into:

"The variability of this upwelling system has been studied on seasonal time scale (Torres, 2003 and Alvarez et al., 2005)"

6) Line 39-41. "The latter" denotes Azores High Pressure? I think ITCZ is also a part of Hadley Circulation system.

It was an error, we apologize for that and we have reformulated this sentence in the new manuscript into:

"In the CUS, the strength of the upwelling favorable winds are associated with latitudinal variation of the Inter-tropical Convergence Zone (ITCZ) and the Azores high pressure system which are both part of the Hadley-circulation. The Azores high pressure migrates from 25°N in late winter and 35°N in late summer."

**7) Line 42. Any reference?**

Thanks, we have added these references below in the new manuscript

Wooster, W.S., Bakun, A., McLain, D., 1976. The seasonal upwelling cycle along the eastern boundary of the North Atlantic. J. Mar. Res. 34 (2), 131–141.

Mittelstaedt, E., 1991. The ocean boundary along the northwest African coast: circulation and oceanographic properties at the sea surface. Prog. Oceanogr. 26, 307–355.

Van Camp, L., Nykjaer, L., Mittelstaedt, E., Schlittenhardt, P., 1991. Upwelling and boundary circulation off northwest Africa as depicted by infrared and visible satellite observations. Prog. Oceanogr. 26, 357–402.

Nykjær, L., Van Camp, L., 1994. Seasonal and interannual variability of coastal upwelling along northwest Africa and Portugal from 1981 to 1991. J. Geophys. Res. 99 (C7), 14197–14207.

Benazzouz, A., Mordane, S., Orbi, A., Chagdali, M., Hilmi, K., Atillah, A., Lluís Pelegrí, J., and Hervé, D.: An improved coastal upwelling index from sea surface temperature using satellite-based approach – The case of the Canary Current upwelling system, Continental Shelf Research, 81, 38–54, https://doi.org/10.1016/j.csr.2014.03.012, 2014.

8) Line 60. Due by => Due to.

Thanks for this remark "Due by" was indeed changed into "Due to" in the new manuscript

9) Lines 70-71. What data did Bakun use for the study? Might be good to describe it.

We thank the reviewer for this suggestion and we have reformulated this sentence in the new version into:

"By using the averages of the meridional wind stress component derived from ship reports, Bakun (1990) suggested that coastal upwelling intensification would occur in response to continued global warming."

10) Line 78. Sea Surface Temperature can be small letter? In Fig.1 I cannot see any black/magenta dots nor any other notifications the caption tells. Probably, forgot to show them in the figure?

Thanks for this remark "Sea Surface Temperature" was changed into "sea surface temperature" and we apologize for the Fig. 1 it was an error, we added the black and magenta dots" on Fig. 1 of the new manuscript.

11) Line 208-209. The action of wind.. should be influence of wind on the upwelling?

Thanks, we have changed these lines in the new manuscript into:

"As described in the introduction section, the influence of wind on the upwelling can be separated into two mechanisms (Sverdrup et al., 1942; Yoshida, 1995 and Smith, 1968)"

12) Definition of Tgeo. I am not familiar with this dynamical parameter to describe the vertical transport due to geostrophic flow. However, when there is SSH meridional gradient, MLD would have also meridional gradient, wouldn't it? Could you please explain why it is ok to use a box-averaged MLD?

We are grateful to the reviewer for this point:

"The cross-shore geostrophic transport (expressed in sverdrup) is computed following this equation:

Tgeo = MLD.g/f (SSHnorth – SSHsouth) where  $\Delta$ SSH= SSHnorth – SSHsouth is the coastal SSH difference between the northern and southern ends of our region of interest and we integrate this transport on the mixing layer by assuming that the geostrophic transport is limited to this layer."

13) Fig. 2, Fig.3 and Fig.4: The panels for the observations (left column) are different-size (also label of latitude) from those for models. I strongly suggest to have same format among them so that it is easier to compare.

The reviewer is right that panels for the observations are different-size (also label of latitude) from those for models. This has been corrected for the Fig.2, Fig. 3 and Fig. 4 in the new manuscript.

14) what does the contour denote?

We are grateful to the reviewer for this remark and we are added in the new manuscript the comment of the contour:

"For Fig. 2: On each panel, the black (grey) contour shows the contour zero (values > 3°C)."

"For Fig.3 and Fig.4: On each panel, the black (grey) contours show the contours  $0.5 \text{ m}^2\text{s}^{-1}$  and  $0.75 \text{ m}^2\text{s}^{-1}$ (values >  $2.5 \text{ m}^2\text{s}^{-1}$ )".

15) Line 275. UISST => UISST ?

Thanks "UISST" was replaced by "UIsst"

16) Line 276. Not clear "negligible of even negative value of CSET".

Sorry it was a mistake. The sentence was corrected in the new manuscript into:

"As for UIsst (Fig. 2), negligible or even negative values of CSET are detected along the IP coast during wintertime indicating the predominance of downwelling conditions".

17) Fig. 4 Along n/s MoUS regions, the upwelling index is almost always negative, indicating downwelling motion is dominant through the whole year. But, this seems contradict against the cool SST there (e.g., Fig.2). So, the cool SST comes from horizontal advection, not from upwelling around Moroccan coast?

We thank the review for the remark, but there is not necessarily a contradiction here. Indeed Fig.4 shows the contribution of Ekman pumping only, which is only one of the dynamical drivers of the upwelling. Over the sMoUS and nMoUS this effect is favorable to a downwelling. This situation result to the negative  $\partial v/\partial x$  (see Eq:4). However, Ekman transport (Fig.3) remains favorable to an upwelling and Fig. 2 confirms that an upwelling is indeed taking place in this regions.

**18) Line 290. null => zero?**

The word "null" was removed and replaced by "zero".

19) Line 305-306. Not clear "sub-regions". Do the authors mean other regions? (IP, nMoUS, sMoUS)?

In these lines sub-regions indeed refers to IP, north and south Morocco.

We have reformulated this sentence into "In the SMUS (Fig B1, panel d) the SSH difference is also always negative and the related amplitude strongly differs from the others sub-regions (IP, nMoUS and sMoUS)".

**20) Line 307-308. Repetitive.**

Thanks, this sentence "the SSH difference is strong all year long in both datasets and tend to be maximum at the beginning of the upwelling season." was removed in the new submitted version.

**21) Fig.5 the xaxis-label data1/2/3 should be AVISO/GODAS/MLD. This can shorten the caption**

Thanks for this remark but the xaxis-label data1/2/3 correspond to the transport total which combined different validation datasets as explained in the caption. Therefore it is not possible to attribute them to a specific data. We thus propose to keep our previous caption in the paper.

22) Fig.5.a => Fig.5a. This expression can be seen elsewhere in the manuscript.

Thanks, this was corrected in several places in the new manuscript.

23) Definition of UItotal. The authors add Ekman transport and pumping to estimate the total upwelling intensity. But this summation doesn't double-count Ekman dynamics (transport and upwelling)? In general, the Ekman pumping compensates the divergence of Ekman transport at the upper level. How do the authors interpret this?

We agree that Ekman suction and coastal divergence are added together but are not really independent in the calculation because they overlap spatially. This point is raised in Jacox et al. (2018) who propose to calculate unambiguously the total divergence associated with Ekman divergence + Ekman suction by integrating the Ekman transport along all boundaries (north, south, east) of the region of interest. The comparison of Jacox et al. (2018) method and the estimation proposed in this submitted manuscript was tested after a similar question from a reviewer of my manuscript Sylla et al 2019 (cited in this manuscript). This comparison (Fig.11) shows that both methodologies in general yield very similar results.In the validation data sets, the difference is less than 5%, with the Jacox's et al. (2018) approach leading to slightly stronger results, while the multimodel mean is weakened by approximately 10%. Given the similarity of these results, and the interest, in our view, to discuss the open ocean wind stress curl separately from the offshore transport divergence, we consider that the overlap is weak and decide to keep the estimation described in the manuscript to compute the Ultotal.

We are added this text below in the new manuscript (line 316-328):

"The physical and biogeochemical responses to coastal divergence and Ekman suction differ in important ways (Capet et al., 2004 and Renault et al., 2016). As a first approach, the CSET and Wek may nevertheless be added up to provide an estimate of upwelling strength. Jacox et al., 2018 have recently suggested that the effect of Ekman processes should be estimated globally from the integration of Ekman transport along the boundaries (north, west, and south) of the region of interest. Comparison of this approach with the one proposed here had been performed with CMIP5 (Sylla et al., 2019). This comparison shows that both methodologies in general yield very similar results. In the validation data sets, the difference is less than 5 %, with the Jacox's et al. (2018) approach leading to slightly stronger results, while the multimodel mean is weakened by approximately 10%. Given the similarity of these results, and the interest, in our view, to discuss the open ocean wind stress curl separately from the offshore transport divergence, we consider that the overlap is weak and decide to estimate the total upwelling intensity (Ultotal) as a sum of the integrated Ekman transport (CSET), the Ekman pumping (Wek) and the geostrophic flow Tgeo. Furthermore, the comparison of this indirect estimate to a more direct estimate from vertical velocities was also done in Sylla et al., 2019 for CMIP5. The authors show that Ultotal is consistent with a direct estimation of the upwelling flux from vertical velocities diagnosed from the models."

24) Section 4. There is no plot of SST itself from MIP models. I am curious how good/bad the SST climatology (seasonal cycle, location of Senegal-Mauritania Front).

Thanks for this remark, we show in Fig.2 the amplitude of the SST seasonal cycle in the climate models. The magnitude of this SST seasonal cycle is maximum over the Senegal-Mauritania region between 12°N and 20°N and 16°W-20°W, because the seasonal upwelling contributes to wintertime cooling. This figure shows that the models in group 2 generally reproduce realistic amplitude in the

correct latitudinal band compared to observation (OISST v2) whereas its intensity is less marked in group 1. Additionally for group 1\* and 2\*, MPI-ESM models generally reproduce an intensified amplitude of the SST seasonal cycle in the Senegal-Mauritania latitude band and in CMCC-CM2 it is amplified only in the northern part of this region. Thus, these bias affect the thermal upwelling indices.

25) Line 341. "mean state of the mean seasonal cycle", climatological seasonal cycle?

Thanks for this remark " the mean seasonal cycle" was indeed changed into "climatological seasonal cycle."

26) Lines 351-55. These under/overestimated upwelling (UISST index) is consistent with the SST bias in each model? As I commented, better to show SST or its bias plot among MIP models.

Thanks again for this remark and as mentioned above (answer of question 24) the bias of UIsst index is consistent with the SST bias in each model (see Fig.2).

27) While I can see the reduced overestimation of IP's upwelling, there might be still some overestimation, especially, in ECMWF-IFS-HR?

The upwelling indices are in general sightly overestimated in the group 2 (HR models) along the Iberian Peninsula (IP) coast. This situation may explain sometimes the higher skill score of group 1 (LR models) described in section 4.4 (Fig. 6, first column) in this sub-region.

28) Line 383. Please remove "in these subdomains"

Thanks, "in these subdomains" was removed.

29) Line2 383-384. "being this...established" might be rephrased?

Thanks, we have changed this sentence in the new version into: "Focusing on the sMoUS and nMoUS, group 1 largely overestimates CSET, whereas this overestimation is less well clear for group 1\* (Fig.3)".

30) Line 385. "validation dataset" is replaced with observations.

Thanks, "validation dataset" was replaced by "observations"

31) Line 386. "Slightly", at least, between Group 1 and 2 the improvement is very

remarkable?

Thanks the word "slightly" was removed.

32) Line 390 and somewhere. "Let's now..." sounds too casual for a scientific paper.

We agree with the reviewer that "Let's now..." sounds too casual for a scientific paper.

We have modified the sentence in the new manuscript into:

"We consider now the ability of the different model configurations to reproduce the seasonal variability of the wind stress curl (Fig.4)". (line 390-391) and line 414- 415: "We consider now the total upwelling transport (UItotal) computed as the sum of all dynamical effects, as explained in section 3.2"

33) Line 391. In Fig.4, it seems that Group 1 models do not have large bias in SMUS region. Like Fig.3. However, Wek and CEST indices are based on wind stress and they may have some coherency, I guess. I am wondering why these indices seem to have different bias in LR models (Group 1).

We agree with the reviewer that the Ekman transport and Ekman pumping are both based on the wind stress and therefore should hold some coherency. However, they also hold differences. For example along the SMUS where the strong difference between CSET and Wek is observed, the zonal component of wind stress is not taken account in *CSET* because the coast is oriented north to south (Eq: 3). Thus the difference between Wek and *CSET* biases could come from this component for example.

34) Line 430. "estimation of the upwelling transport", sounds a bit strange in this context.

The estimation of the upwelling transport used here are able to fully capture the estimation of the upwelling transport" -> "estimate quite realistically the upwelling transport.

35) Line 432-433. "goes in the...UItotal", rephrase.

We thank the reviewer for this suggestion.

These lines ware corrected in the new manuscript into" Groups 1\* and 2\* show similar range of Ultotal and no clear effects due to the increasing resolution are identified.

36) Line 473-480. This part is a bit redundant

Thanks for this remark, this part was removed in the new manuscript.

37) Line 488-489. Rephrase.

Thanks, this line was changed into:

**"Globally, our results show that observations and reanalyses yielding a fairly consistent picture of the CUS climatology.**

Anonymous Referee #2: Review of Manuscript gmd-2022-130

**Summary:**

Authors investigate the realism of Canary upwelling system simulation in 6 high-resolution & standard resolution global coupled climate models from the HighResMIP project. Upwelling indices based on sea surface temperature (SST), height (SSH), and surface wind stress during the 1985-2014 period have been analyzed from the models and compared against that from observations. Authors find that increasing spatial resolution of atmosphere and ocean components of coupled models improves upwelling simulation only in the southern part of the upwelling system and while worsening it in the northern part.

The topic addressed is very relevant, since it is crucial to understand the upwelling dynamics in the Canary upwelling system (CUS) in a coupled framework at a high- resolution to better address future climate change and associated societal impacts. However, after considering the scientific merit, analysis methods, novelty, and overall presentation, I have a few major comments as detailed below.

We thank the reviewer for his general comment. We answer below each of his/her points.

**Major comment**

1) The essential background details for this study are not presented in a clear manner. What resolutions (for the ocean and atmospheric components) are considered standard and high resolutions? According to Chelton et al. 1998), the first baroclinic Rossby radius of deformation varies in the range of 20-60 km and the high-resolution ocean models in this study (0.250, Table 1) can barely resolve this scale in most parts of the CUS. Most of the high-resolution atmospheric components in this study are about 0.50 which may not be able to resolve realistic wind structure/drop-off near the coast (see Patricola and Chang, 2017). So, even though 0.25 deg ocean model and 0.5 deg atmospheric model are described as "high-resolution" in this study, it is not shown that these resolutions/models can realistically resolve upwelling dynamics in this region (also see comment 2 below). No discussions/insights are offered about possible mechanisms/processes which are not resolved at these resolutions, compared to typical regional model resolutions of 0.10 in the ocean and 0.250 in the atmosphere.

Indeed, the reviewer is right and we are added this text below in the new manuscript (line 145-152).

Furthermore "standard" and "high resolution" terms are rather subjective and depend on the context. Here, we use them in the context of global climate modelling, so that standard resolution is around 1° for the ocean, and high-resolution around 0.25°. We acknowledge that the high-resolution ocean models in this study can barely resolve the first baroclinic Rossby radius deformation (20-60 km, Chelton et al 1998) in most parts of the CUS. Similarly, the standard atmospheric resolution is 1° to 2.5° while most of the high-resolution atmospheric components in this study are about 0.5° which may not be able to resolve realistic wind structure/drop-off near the coast (Patricola and Chang, 2017). So, even models described here as "high-resolution" can probably realistically not resolve upwelling dynamics in this region, at least not as well as dedicated configurations (for example ROMS numerical simulation including a high resolution grid  $1/60^{\circ}$  (~2 km) and a standard resolution  $1/12^{\circ}$  (~10 km); e.g. Ndoye et al., 2017)

2) Lack of analysis/discussion of mean upwelling vertical structure (eg. temperature depth- distance sections, see Fig.5 in Capet et al. (2004) and coastal wind structure (eg. wind profiles, see Fig.1 in Capet et al. (2004)) against observations makes it difficult to evaluate the realism of modeled winds and upwelling. The presented seasonal cycle of upwelling indices alone does not help in this regard. How realistic is the coastal wind drop- off (see Capet et al., 2004) in the 0.50 atmospheric model compared to that in the observations? How realistic is the vertical structure of temperature in terms of up-sloping isotherms? How do the high-resolution models differ from low-resolution ones in these aspects?

We agree with the reviewer that the mean modeled oceanic and atmospheric states in the upwelling region have not been precisely qualified in this present study. This point is also underlined in the final manuscript. The focus was on upwelling indices, and we leave it to other studies to link the performance highlighted here to the climatology.

3) The upwelling zone definition (using rectangular regions, especially in nMoUS and sMoUS regions) for analysis is not consistent with the narrow-coastal upwelling pattern (Fig.1, blue box). For example, at about 31oN in the nMoUS region, the coastal zone stretches about 8-90s including the non-upwelling offshore region. At 21oN, it reduces to about 1-20s width. Hence this approach is not consistent, especially for comparing different regions like nMoUS and sMoUS (especially for fields like wind stress curl & models with low-resolution). A fixed-width approach like that in Jacox et al. (2018) (see their Fig.1) will be better suited here.

We thank the reviewer for this remark, we have indeed changed the box in the Morocco region (see Fig. 1 in new manuscript). However the results (see Fig.4, Fig.5.b and Fig.5.c) are very similar to what we have submitted in the previous manuscript. Nevertheless we thus propose to keep this new method in the paper.

4) The estimation of total upwelling intensity (lines 295-296) by simply adding three indices (measuring Ekman transport, Ekman pumping, and geostrophic transport) is not convincing since it is not verified in any manner (say against vertical velocity from the model). Please note that Jacox et al. (2018) (cited in this manuscript) compute the total upwelling index/transport without considering Ekman pumping explicitly (but including its effect by integrating Ekman transport around the perimeter of coastal boxes) and shows that it matches very well with the transport estimated from model's vertical velocity. Such verification is required for the method used here.

We agree that we do not use exactly the same methodology as in Jacox et al. 2018. Nevertheless, the comparison to the latter had been performed with CMIP5 data for the SMUS only and he shown again here (not published, Fig. 1). Furthermore, the comparison of this indirect estimate to a more direct

estimate from vertical velocities was done in Sylla et al 2019 for CMIP5. Main conclusions are now added in this manuscript. Finally, we emphasize the fact that the point of this study is not necessarily to come up with a quantitative assessment of the upwelling but rather to compare various quantitative assessments against resolution. This point is made clearer when estimation of total upwelling intensity is introduced in the new manuscript (text below)

"The physical and biogeochemical responses to coastal divergence and Ekman suction differ in important ways (Capet et al., 2004 and Renault et al., 2016). As a first approach, the CSET and Wek) may nevertheless be added up to provide an estimate of upwelling strength. Jacox et al., 2018 have recently suggested that the effect of Ekman processes should be estimated globally from the integration of Ekman transport along the boundaries (north, west, and south) of the region of interest. Comparison of this approach with the one proposed here had been performed with CMIP5 (Sylla et al., 2019). This comparison shows that both methodologies in general yield very similar results. In the validation data sets, the difference is less than 5 %, with the Jacox's et al. (2018) approach leading to slightly stronger results, while the multimodel mean is weakened by approximately 10%. Given the similarity of these results, and the interest, in our view, to discuss the open ocean wind stress curl separately from the offshore transport divergence, we consider that the overlap is weak and decide to estimate the total upwelling intensity (Ultotal) as a sum of the integrated Ekman transport (CSET), the Ekman pumping (Wek) and the geostrophic flow Tgeo. Furthermore, the comparison of this indirect estimate to a more direct estimate from vertical velocities was also done in Sylla et al., 2019 for CMIP5. The authors show that Ultotal is consistent with a direct estimation of the upwelling flux from vertical velocities diagnosed from the models."

5) The available resolution of models (~0.50 to 2.50 in the atmosphere and 0.250 to 10 in the ocean) and the combination of coarse and high-resolution atmospheric and ocean components in this study are not sufficient to draw the conclusion that high-resolution in the atmosphere "has only a limited impact" (eg. lines 456-457) in a general sense. Only a comparison of a high-resolution ocean grid (~10 km to resolve the first baroclinic Rossby radius well) with coarse (~10) and fine (~1/40) atmospheric grids can isolate the true impact of a high-resolution in the atmosphere. In other words, the ocean resolution should be fine enough to fully utilize the well-resolved coastal wind drop-off (see Capet et al. (2004) and Patricola and Chang (2017)).

We agree with the reviewer that the range of resolution that is explored here is rather limited. This study is based on an ensemble of opportunity of coupled climate models, which furthermore includes quite a limited number of models and configurations.

We have leveled down the conclusions in the new manuscript (line 468) by adding systematically "within the investigated range"

**Technical Corrections**

6)- Please explain the analysis methods in detail (eg. definition of the seasonal cycle, integration steps to compute total upwelling intensity etc.)

We use monthly climatologies built over the 1985–2014 time period for the climate simulations and observations. Thanks

7) Line 6-7: The sentence "Our analysis shows that an increase of spatial resolution depends on the sub-domain of the CUS considered." is ambiguous. It should be "...shows that an improvement in upwelling simulation due to the increased spatial resolution....". -

Thanks, we are changed this sentence into:

"Ours analysis shows that possible improvement in upwelling representation due to the increased spatial resolution depends on the sub-domain of CUS considered."

8)Line 8: "both components": Though it is mentioned that the models are coupled, explicitly state "both atmosphere and ocean components" for clarity.

Both components mean here the resolution of the ocean and atmosphere model. Thanks

9)Line 25: Please cite Capet et al. (2004) for the role of coastal wind drop-off in wind stress curl-driven upwelling.

We are grateful to the reviewer for pointing out this paper.

We now cite this paper as a reference for the role of coastal wind drop-off in wind stress curl-driven upwelling (line 28 in the new version)

10) Fig.1: Black and magenta stars and dots are mentioned in the caption but are not visible even after trying different PDF viewers. It will be helpful to overlay a few SST contours for highlighting the cooler SSTs in the upwelling region

Thanks for this remark we added on Fig.1 of the new manuscript version the black and magenta dots and also the SST contours for highlighting the cooler SSTs in the upwelling region.

11) Also, show the region over which Ekman pumping has been integrated (line 224).

The Ekman pumping was integrated over the latitude and longitude range of each box on Fig. 1 for each sub-regions.

12) Table 2: Use "reanalysis" instead of "reanalyse".

Thanks, "reanalyse" was changed into "reanalysis"

13) - Fig.1 & 2: Technical inconsistency: Fig.1: Various regions extend from 12N to 42N, with a blue dashed line representing the northern boundary of sMoUS region.

The various regions of the CUS are located from the Senegalese coast (12°N) to the Iberian Peninsula region (43°N). The different regions of CUS are also represented of the different boxes as explained in the caption of Fig.1. The Moroccan is separated into two sub-domain based on the seasonality of upwelling (nMoUS: 26°N 32°N and the sMoUS: 21°N-25°N). This division of the Morocco system was correctly represented on Fig.1 in the new manuscript, thanks.

14) Fig.2: Some of the panels do not extend to 12°N and now the blue dashed line represents the northern boundary of nMoUS region.

Thanks for this remark, but all of the panels in Fig.2 are extended to 12°N in the previous submitted manuscript. This remark that some of the panels do not extend to 12°N in the previously submitted manuscript is due to the fact that the panels for the observations (left column) were different-size from those for models in the subplot. It was correct this mistake and in the new manuscript all panels are the same format and the blue dashed lines consistently represents the nMoUS sub-region.

15) On line 259, explicitly state "observations and reanalysis" as in line 272.

Thanks we are added "observations and reanalysis" in line 259.

16) Also, which all SST values are contoured in Fig.1? (difficult to read from the colorbar). The dark-red colors in panels (eg. last column,2nd row) is not seen in the colorbar.

We are grateful to the reviewer for this remark and we are added on Fig.1 in the new manuscript some SST contours for highlighting the cooler SSTs in the upwelling region. For Fig. 2: on each panel we are added black (grey) contour that shows the contour zero (values > 3°C). For Fig.3 and Fig.4 we are added also black (grey) contours that show the contours  $0.5m^2s^{-1}$  and  $0.75m^2s^{-1}$ (values >  $2.5m^2s^{-1}$ ).

17) - Fig.B1: Colors do not have any correspondence to positive/negative values. Make the color scale from -0.14 to 0.14

Thanks for this remark, we are changed the scale color from -0.14 to 0.14 in the new manuscript version.

18)- Line 343: Fig.2: CMCC-CM2 (Group 1\* and 2\*) still shows a high UI\_sst index in the summer, though the sign of UI\_sst is positive throughout the year.

The reviewer is right that the Line 343 was biased because CMCC-CM2 shows a high and positive UIsst values in summer over the Iberian Peninsula. This line was removed.

However the sign of UIsst is positive throughout the year inCMCC-CM2 models. But it is also important to keep in mind that in this region the upwelling occurs generally in summer as in the previous studies referenced in the introduction of the submitted manuscript. Therefore our analysis was concerning in this period. Additionally this index was used, but present disadvantages. In general, the main disadvantage of UIsst comes from the fact that changes in coastal and oceanic temperature cannot always be attributable to upwelling. Indeed UIsst can be strongly influenced both by local scale phenomena (e.g., the presence of rivers with high runoff which can modify the SST signal near the coast) and by macroscale phenomena (e.g., El Nino can result in changes in coastal temperature that are not related to the presence of coastal upwelling, eg Gomez et al 2008).

19)- Lines 347-349: For both groups 1\* and 2\*, upwelling is present in the nMoUS region indicated by positive values. But the pattern is not the same as in the observations.

The reviewer is right that the Lines 347-349: was not correct, this has been changed into: "For both groups 1\* and 2\*, the upwelling is broadly reproduced in these subregions, with an overestimation of UIsst amplitude in MPI-ESM1 over the sMoUS."

20)- Line 349: increasing just the atmospheric resolution makes the summer upwelling stronger in the IP region in MPI-ESM1-2 case. This is against the statement in line 365 too.

Thanks for this remark, however the comparison of group 1\* and group 2\* does not show very clear difference of UIsst amplitude during summer along the IP sub-region (Fig.2) excepted in MPI-ESM. Therefore our analysis in line 349: "Thus, the only increase of the atmospheric resolution in models produces no clear impact on upwelling representation" is similar to what we have noted in line 365: "Thus, the IP does not seem to be very sensitive to these changes in model resolution." Indeed these two lines explain that there are no clear improvement to increasing only atmosphere or both component (ocean and atmosphere) resolution in this sub-region.

-21) Section 3.1 title (Line 338): Change it to "The thermal upwelling indices"

Thanks, "The thermal upwelling" was changed into: "The thermal upwelling indices".

22) Section 3: All figures referred in this section have panels from both observation and models, but figures from models are discussed only in Section.4. The title for section 3 alone is not sufficient to bring this point to readers' attention.

This is true. Nevertheless, to make the manuscript lighter and easier for the reader, we have decided to make these general plots but describe them sequentially in 2 sections. Additionally in each section we noted clearly the position of the observation datasets or models we are analyzing.

We add nevertheless this text in the new manuscript (section 3):

"In this section, we describe the upwelling indices defined above computed for the observation datasets. These indices are shown in Fig 2, Fig.3, Fig.4 and Fig.5 for both the data and models but only the observations panels are described here. The results from the modeling experiments will be described in section 4".

23) line 192: Need to provide a basic definition of MLD criteria/method in addition to the reference.

Thanks, we added this text in below in the new manuscript:

"This MLD climatology is based on ARGO profiles where MLD was estimated following a density criterion at a monthly resolution. The selected criterion is a threshold value of temperature (namely 0.2 °C) from a near-surface value at 10 m depth."

24) line 299-300: "We have examined....." edit this sentence for clarity.

Thanks, we are changed the line 299-300 into:

"We have examined firstly the monthly climatology of the meridional sea surface height gradient from the AVISO satellite data and the GODAS reanalysis (see first two columns in Figure B1 of the appendix B)".

**Anonymous Referee #3: Review of Manuscript gmd-2022-130**

**Summary:**

Authors investigate the Canary upwelling system with the use of six global climate models at high and standard resolution from the HighResMIP project covering quite a long period of time from 1950-2014. The analysis done by the authors is based not only on the SST, wind stress, sea surface height, but also in the layer depth fields. In that sense, I would like to arise my minor comment or just a curiosity.

We thank the reviewer for his general comment and appreciation of our manuscript. We answer below each of his/her points.

**Minor comment:**

1.- Bonino et al. (2019) pointed out that both the wind stress and the stratification should be considered in order to evaluate future changes in coastal upwelling. Bakun (1990) did not consider the stratification and other processes that might change the thermocline depth such as coastal trapped waves. Warming in coastal areas increases the stratification and inhibits the vertical nutrient exchange limiting the productivity (Brady et al., 2019; Di Lorenzo et al., 2005; Garcia-Reyes et al., 2015). Coastal trapped waves can also change the water column stratification and cause anomalies affecting the productivity (Bachèlery et al., 2016; Echevin et al., 2014; Pietri et al., 2014; Rykaczewski & Dunne, 2010). In fact, both the wind stress and the stratification are able to amplify or mitigate the upwelling intensity.

Consequently, changes in wind stress and stratification might be complementary or competitive for the upwelling intensity in a global warming scenario (Siemer et al., 2021; Bachèlery et al., 2016; Pietri et al., 2014). Moreover, Bonino et al. (2019) and recently showed by Siemer at al., 2021, found that in the Canary upwelling system the stratification and coastal trapped waves seem to have stronger effects on the upwelling intensity than in other EBUS such as the Benguela system where a positive linear relationship exists between the upwelling intensity and the wind stress.

Do you think that the improve in the global model performance is exclusively based on the increased in resolution or other physical parameters could be responsible of this improvements that are include in the HR models instead of the LR?? Have you investigate this complementary or competitive factor for the upwelling intensity due to the wind stress and stratification in the CUS?

We would like to thank the reviewer for this detailed and interesting comment. We were not aware of all these references in details, in particular not Siemer et al 2021. The effect of stratification is indeed not investigated in this study. Comparing ocean stratification and vertical transport between groups 1 and 2 in particular can indeed provide insight into the relative role of increased ocean and atmospheric resolution in improving the representation of upwelling.

We have thus added this discussion at the end of the paper as a perspective (text below):

"The effect of stratification in particular is not investigated in this study. Comparing ocean stratification and vertical transport between groups 1 and 2 can indeed provide insight into the relative role of increased ocean and atmospheric resolution in improving the representation of upwelling."

**References**

Capet, X. J., Marchesiello, P., and McWilliams, J. C. (2004), Upwelling response to coastal wind profiles, Geophys. Res. Lett., 31, L13311, doi:10.1029/2004GL020123.

Chelton, D. B., R. A. deSzoeke, M. G. Schlax, K. E. Naggar, and N. Siwertz (1998), Geographical variability of the firstbaroclinic rossby radius of deformation, J. Phys. Oceanogr., 28, 433–460.

Patricola CM, Chang P (2017) Structure and Dynamics of the Benguela Low-Level Coastal Jet. Climate Dynamics, 49, 2765-2788.

Siemer, J. P., Machín, F., González-Vega, A., Arrieta, J. M., Gutiérrez-Guerra, M. A., Pérez-Hernández, M.D., Vélez-Belchí, P., Hernández-Guerra, A. and Fraile-Nuez, E.

(2021). Recent trends in SST, Chl-a, productivity and wind stress in upwelling and open ocean areas in the upper Eastern North Atlantic subtropical gyre. Journal of Geophysical Research: Oceans, 126, e2021JC017268. https://doi.org/10.1029/2021JC017268

Bonino, G., Di Lorenzo, E., Masina, S. & Iovino, D. (2019). Interannual to decadal variability within and across the major eastern boundary upwelling systems. Scientific Reports, 9, 19949. https://doi.org/10.1038/s41598-019-56514-8

**List of all relevant changes made in the manuscript:**

Line 2: The word "operating" was replaced by "operated" in the new manuscript.

**Abstract:** 'For this project the resolution of the ocean/or atmosphere components was increased.' this sentence was removed in the new manuscript,

Lines 18. References below were added in the new manuscript:

Herbland, A., Voituriez, B., 1974. La production primaire dans l'upwelling maur- itanien en mars 1973. Cah. O.R.ST.OM., Sér. Océanogr. 12 (3), 187–201.

Minas, H.J., Codispoti, L.A., Dugdale, R.C., 1982. Nutrients and primary production in the upwelling region off Northwest Africa. Rapp. P.-V. Reun., Cons. Int. Explor. Mer 180, 148–183.

*Tretkoff, E. (2011). Research Spotlight: Coastal cooling and marine productivity increasing off Peru. Eos Transact. Am. Geophys. Union 92, 184–184. doi: 10. 1029/2011eo210009*

Huyer, A. (1983). Coastal upwelling in the California Current system. Prog. Oceanogr. 12, 259–284. doi: 10.1016/0079-6611(83)90010-1

Line 25: The word "positive" was removed and replaced by "cyclonic" in the new manuscript.

Line 34-35. This sentence as reformulated in the new manuscript into:

"The variability of this upwelling system has been studied on seasonal time scale (Torres, 2003 and Alvarez et al., 2005)".

**Line 36-39** were changed in the new version into: "In the CUS, the strength of the upwelling favorable winds are associated with latitudinal variation of the Inter-tropical Convergence Zone (ITCZ) and the Azores high pressure system which are both part of the Hadley-circulation. The Azores high pressure migrates from 25°N in late winter and 35°N in late summer."

Line 40: References below were added:

Wooster, W.S., Bakun, A., McLain, D., 1976. The seasonal upwelling cycle along the eastern boundary of the North Atlantic. J. Mar. Res. 34 (2), 131–141.

Mittelstaedt, E., 1991. The ocean boundary along the northwest African coast: circulation and oceanographic properties at the sea surface. Prog. Oceanogr. 26, 307–355.

Van Camp, L., Nykjaer, L., Mittelstaedt, E., Schlittenhardt, P., 1991. Upwelling and boundary circulation off northwest Africa as depicted by infrared and visible satellite observations. Prog. Oceanogr. 26, 357–402.

Nykjær, L., Van Camp, L., 1994. Seasonal and interannual variability of coastal upwelling along northwest Africa and Portugal from 1981 to 1991. J. Geophys. Res. 99 (C7), 14197–14207.

Benazzouz, A., Mordane, S., Orbi, A., Chagdali, M., Hilmi, K., Atillah, A., Lluís Pelegrí, J., and Hervé, D.: An improved coastal upwelling index from sea surface temperature using satellite-based approach – The case of the Canary Current upwelling system, Continental Shelf Research, 81, 38–54, https://doi.org/10.1016/j.csr.2014.03.012, 2014.

Line 58: "Due by" was indeed changed into "Due to".

Line 68:69 was changed in the revised version into: "By using the averages of the meridional wind stress component derived from ship reports, Bakun (1990) suggested that coastal upwelling intensification would occur in response to continued global warming."

Line 76: "Sea Surface Temperature" was changed into "sea surface temperature"

Fig. 1: we added the black and magenta dots".

Line 205: was modified in the revised version into "As described in the introduction section, the influence of wind on the upwelling can be separated into two mechanisms (Sverdrup et al., 1942; Yoshida, 1995 and Smith, 1968)".

Fig2, Fig3 and Fig. 4: Panels for the observations and models are now the same size in the new manuscript.

Comment of contours were added on Fig. 2, Fig.3 and Fig. 4 on the new manuscript.

Line 270: "UISST" was replaced by "UIsst"

Line 271- 272: we are changed this sentence into: "As for UIsst (Fig. 2), negligible or even negative values of CSET are detected along the IP coast during wintertime indicating the predominance of downwelling conditions".

Line 285: The word "null" was removed and replaced by "zero".

Line 300-301: we are modified this sentence in the revised version into: "In the SMUS (Fig B1, panel d) the SSH difference is also always negative and the related amplitude strongly differs from the others sub-regions (IP, nMoUS and sMoUS)".

**In section 3.3**: This sentence: "The SSH difference is strong all year long in both datasets and tend to be maximum at the beginning of the upwelling season." was removed in the new submitted version.

Fig.5.a was replaced by Fig.5a in several places in the new manuscript.

Paragraph line 308-320 was added in the new version.

Line 333: the mean seasonal cycle" was indeed changed into "climatological seasonal cycle."

Line 373: The word in these subdomains" was removed.

Line 374: "validation dataset" was replaced by "observations"

line 375: the word "slightly" was removed.

Line 379 and line 403 "Let's now..." was replaced by "we consider now" in the new manuscript.

**Line 420-421:** we are reformulated this sentence in the revised version into: "Groups 1\* and 2\* show similar range of UItotal and no clear effects due to the increasing resolution are identified.

**In Section 5 this paragraph below was removed in the new manuscript. It was considered by the reviewer as a bit redundant:**

Climate models provide, by construction, imperfect representation of the climate system. In particular, and because of their coarse resolution, the performance of global climate models to simulate the coastal upwelling systems is subject of a wide discussion. The biases of the climate models with both an oceanic and/or atmospheric origin are closely linked to limitations in the model physic formulation and insufficient model resolution (Li and Xie, 2012; Zuidema et al., 2016 and Harlaß et al., 2018). Nevertheless, and are useful to assess, it has been shown that coupled models (CMIP5/CMIP6) are able to reproduce some features of these systems and are used to assess changes in the future (Wang et al., 2015 and Sylla et al., 2019). The upwelling phenomenon is one of the physical processes most sensitive to the model resolution and for which an improvement is expected when the resolution is increased (Small et al., 2015 and Vazquez et al., 2019).

**Line 468:** was modified into: "Globally, our results show that observations and reanalyses yielding a fairly consistent picture of the CUS climatology".

Paragraph line 141-149 was added in the new manuscript.

Paragraph line 308-320 was added in the new version.

Line 456: We have leveled down the conclusions in the new manuscript by adding systematically "within the investigated range"

Line 5-6: was modified in the revised version into: "Ours analysis shows that possible improvement in upwelling representation due to the increased spatial resolution depends on the sub-domain of CUS considered."

**Line 25:** we are added this reference: Capet, X. J., Marchesiello, P., and McWilliams, J. C.: Upwelling response to coastal wind profiles, Geophysical Research Letters, 31, n/a–n/a, https://doi.org/10.1029/2004gl020123, 2004.

 Table 2: the word "reanalyse" was changed into "reanalysis"

Line 254: we are added "observations and reanalysis".

Fig.B1: we are changed the scale color from -0.14 to 0.14 in the new manuscript version.

Line 338-339:we are modified this sentence in the new version into: "For both groups 1\* and 2\*, the upwelling is broadly reproduced in these subregions, with an overestimation of UIsst amplitude in MPI-ESM1 over the sMoUS."

Section 3.1: The thermal upwelling" was changed into: "The thermal upwelling indices".

**In section 3** we are added this paragraph: "In this section, we describe the upwelling indices defined above computed for the observation datasets. These indices are shown in Fig 2, Fig.3, Fig.4 and Fig.5 for both the data and models but only the observations panels are described here. The results from the modeling experiments will be described in section 4".

Line 189-191 were added in the new version.

Line 294-296 was modified into: "We have examined firstly the monthly climatology of the meridional sea surface height gradient from the AVISO satellite data and the GODAS reanalysis (see first two columns in Figure B1 of the appendix B)".

Line 494-496 were added in the new version.